# Autoregressive Visual Decoding from EEG Signals

**Sicheng Dai**[1,2,3,4]**, Hongwang Xiao**[4,5]**, Shan Yu**[1,3]**, Qiwei Ye**[4*]
[1]Institute of Automation, Chinese Academy of Sciences
[2]School of Artifcial Intelligence, University of Chinese Academy of Sciences
[3]State Key Laboratory of Brain Cognition and Brain-inspired Intelligence Technology
[4]Beijing Academy of Artificial Intelligence
[5]National Key Laboratory for Multimedia Information Processing, Peking University
daisicheng2023@ia.ac.cn, shan.yu@nlpr.ia.ac.cn, qwye@baai.ac.cn

## Abstract

Electroencephalogram (EEG) signals have become a popular medium for decoding visual information due to their cost-effectiveness and high temporal resolution. However, current approaches face significant challenges in bridging the modality gap between EEG and image data. These methods typically rely on complex adaptation processes involving multiple stages, making it hard to maintain consistency and manage compounding errors. Furthermore, the computational overhead imposed by large-scale diffusion models limit their practicality in real-world brain-computer interface (BCI) applications. In this work, we present AVDE, a lightweight and efficient framework for visual decoding from EEG signals. First, we leverage LaBraM, a pre-trained EEG model, and fine-tune it via contrastive learning to align EEG and image representations. Second, we adopt an autoregressive generative framework based on a "next-scale prediction" strategy: images are encoded into multi-scale token maps using a pre-trained VQ-VAE, and a transformer is trained to autoregressively predict finer-scale tokens starting from EEG embeddings as the coarsest representation. This design enables coherent generation while preserving a direct connection between the input EEG signals and the reconstructed images. Experiments on two datasets show that AVDE outperforms previous state-of-the-art methods in both image retrieval and reconstruction tasks, while using only 10% of the parameters. In addition, visualization of intermediate outputs shows that the generative process of AVDE reflects the hierarchical nature of human visual perception. These results highlight the potential of autoregressive models as efficient and interpretable tools for practical BCI applications. Code is available at https://github.com/ddicee/avde.

## 1 Introduction

How can we access and interpret the rich visual information encoded in human brain activity? This question has captivated neuroscientists for decades, driving fundamental research at the intersection of cognitive science and artificial intelligence. Decoding human vision from non-invasive neural signals not only advances our understanding of neural representation mechanisms but also promises transformative applications in brain-computer interfaces. Early pioneering work (Kay et al., 2008; Miyawaki et al., 2008; Naselaris et al., 2009) established that simple visual patterns could be decoded from functional magnetic resonance imaging (fMRI), while recent advances in generative AI have enabled reconstruction of remarkably detailed visual content from brain signals (Takagi & Nishimoto, 2023a; Scotti et al., 2023; Fang et al., 2023).

Despite these successes, fMRI-based approaches face fundamental limitations for practical applications: they operate at temporal resolutions orders of magnitude slower than actual neural processing, require costly infrastructure, and confine subjects to restrictive scanner environments (Menon & Kim, 1999; Logothetis, 2008). These constraints have motivated a shift toward electroencephalography (EEG) for visual decoding (Cichy & Pantazis, 2017). EEG offers millisecond-level temporal

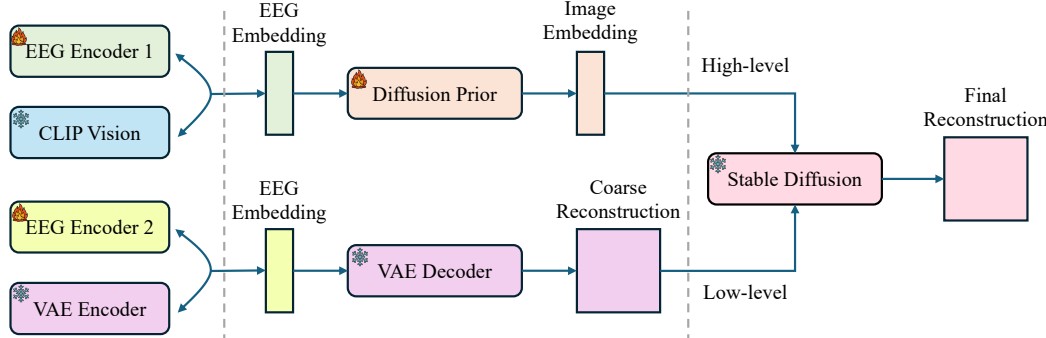

Figure 1: A typical unCLIP framework employed in previous EEG-based visual decoding works Li et al. (2024); Zhang et al. (2025); Xiao et al. (2025); Scotti et al. (2023). Despite its flexibility, the framework comprises multiple stages (five in this case), each introducing potential sources of error that can accumulate and degrade overall performance. Furthermore, the computational and memory demands of its components present significant challenges for practical implementation in BCIs.

precision, while providing significantly greater portability and accessibility at a fraction of the cost. Recent work in EEG-based visual decoding (Li et al., 2024; Xiao et al., 2025; Zhang et al., 2025) has demonstrated promising capabilities in both image retrieval and reconstruction, suggesting the potential for more deployable applications.

However, a fundamental challenge persists: how to effectively bridge the vast distributional gap between noisy EEG signals and structured visual content. This challenge manifests in three limitations of current approaches. First, these methods typically rely on diffusion models and complex, multi-stage adaptation processes based on unCLIP (Ramesh et al., 2022) to project EEG signals into compatible latent representations. The sequential nature of these pipelines inherently compounds errors across stages (Li & van der Schaar, 2023), degrading the fidelity of reconstructed images. Second, the EEG encoders are typically trained from scratch using a limited number of image-EEG pairs, which raises concerns about their capability to capture the intricate features in highly noisy EEG signals. Third, the computational demands of large-scale diffusion models (often exceeding 3B parameters) render these systems impractical for BCI applications where efficiency and responsiveness are crucial.

To address these limitations, we propose Autoregressive Visual Decoding from EEG signals (AVDE), a lightweight and efficient two-stage pipeline for EEG-to-image translation. Our approach makes two key innovations: First, rather than training EEG encoders from scratch, we leverage LaBraM (Jiang et al., 2024)—a model pre-trained on thousands of hours of diverse EEG data—and fine-tune it using contrastive learning to align EEG and image representations. This transfer learning approach substantially improves the extraction of meaningful features from noisy EEG signals. Second, we replace complex multi-stage diffusion processes with a streamlined autoregressive framework based on "next-scale prediction." Our approach encodes images into multi-scale token maps using a pre-trained VQ-VAE (Tian et al., 2024), then trains a transformer to progressively predict increasingly detailed visual representations, starting from EEG embeddings as the coarsest representation. This approach ensures coherent generation while maintaining a direct relationship between EEG signals and visual outputs. Experiments on two datasets demonstrate that AVDE achieves state-of-the-art performance in both retrieval and reconstruction tasks while using only 10% of the parameters required by previous methods. Furthermore, visualization of the intermediate outputs shows that the generative process of AVDE reflects the hierarchical nature of human visual perception, underscoring the potential of autoregressive models as tools for exploring the dynamics of human visual cognition.

In summary, the main contributions are as follows:

- We introduce AVDE, a novel framework for EEG-based visual decoding that employs a hierarchical "next-scale prediction" strategy within an autoregressive transformer. This approach progressively constructs visual representations from coarse to fine details, mirroring

the hierarchical nature of both biological visual processing and computational vision systems.

- We demonstrate that transfer learning from pre-trained EEG model significantly improves visual decoding performance. By fine-tuning the LaBraM encoder (Jiang et al., 2024) with contrastive learning, we achieve more robust alignment between EEG and image representation spaces compared to training EEG encoders from scratch.

- We demonstrate through comprehensive experiments that AVDE achieves state-of-the-art performance in both image retrieval and reconstruction tasks on two datasets, while being more lightweight and computationally efficient than prior methods. Our approach reduces parameter count by approximately 90% compared to diffusion-based methods, making it more suitable for practical BCI applications.

## 2 METHOD

### 2.1 EEG ENCODING WITH LABRAM

A critical challenge in EEG-based visual decoding is extracting meaningful features from the inherently noisy signals. Rather than training encoders from scratch on limited EEG-image pairs, we build upon LaBraM (Jiang et al., 2024), a model pre-trained on over 2000 hours of diverse EEG data spanning multiple datasets and recording conditions.

The architecture processes input EEG data $X \in \mathbb{R}^{C \times T}$ (where $C$ represents channels and $T$ represents time points) through the following encoding scheme:

1) **Temporal patching**: The input EEG signal is segmented in the temporal dimension with a non-overlapping window of length $w$, resulting in patches:

$$\mathbf{x} = \{x_{c_j,k} \in \mathbb{R}^w \mid j = 1, 2, \ldots, C, k = 1, 2, \ldots, \lfloor \frac{T}{w} \rfloor\} \tag{1}$$

2) **Local feature extraction**: Each patch is processed by a temporal encoder comprising stacked convolutional blocks (1D convolution, group normalization, GELU activation) to capture fine-grained temporal patterns:

$$\{e_{c_j,k} \in \mathbb{R}^d \mid j = 1, 2, \ldots, C, k = 1, 2, \ldots, \lfloor \frac{T}{w} \rfloor\} \tag{2}$$

where $d$ is the embedding dimension.

3) **Spatiotemporal contextualization**: To incorporate both temporal and spatial context into the model, we set up two sets of trainable positional embeddings: a temporal embedding set $TE = \{te_k \mid k = 1, 2, \ldots, \lfloor \frac{T}{w} \rfloor\}$ and a spatial embedding set $SE = \{se_j \mid j = 1, 2, \ldots, C\}$. The final patch representation is obtained by summing the corresponding temporal and spatial embeddings with the encoder output:

$$\{e_{c_j,k} + te_k + se_j \mid j = 1, 2, \ldots, C, k = 1, 2, \ldots, \lfloor \frac{T}{w} \rfloor\} \tag{3}$$

4) **Global integration**: The enriched patch embeddings are processed by a Transformer encoder (Vaswani et al., 2017) that models dependencies across both time and channels, effectively integrating information from the entire EEG epoch.

### 2.2 REPRESENTATION ALIGNMENT THROUGH CONTRASTIVE LEARNING

While pre-training provides a strong foundation for EEG feature extraction, the LaBraM model was primarily trained on clinical data (Obeid & Picone, 2016) rather than EEG responses to visual stimuli. To adapt the model for visual decoding, we fine-tune it through contrastive learning, which creates alignment between EEG and image representation spaces.

Given paired EEG-image data ($\mathbf{X} \in \mathbb{R}^{B \times C \times T}, \mathbf{I} \in \mathbb{R}^{B \times H \times W}$), we encode EEG signals using the LaBraM model and images using a frozen CLIP (Radford et al., 2021) encoder, producing embeddings $\mathbf{e}, \mathbf{z} \in \mathbb{R}^{B \times d}$. We then optimize a bidirectional contrastive objective that maximizes agreement between corresponding EEG-image pairs while minimizing similarity between non-corresponding pairs:

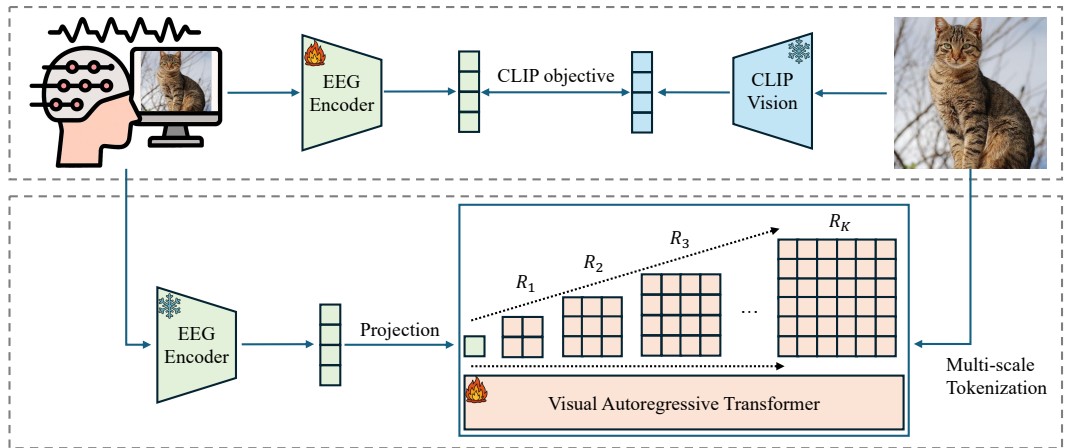

Figure 2: AVDE involves two training stages. **Stage 1:** A pre-trained EEG encoder is fine-tuned using contrastive learning to more effectively capture visual information embedded in EEG signals. This adaptation aims to provide a more informative initialization for the subsequent visual reconstruction process. **Stage 2:** A visual autoregressive transformer is trained using the next-scale prediction objective (Equation 7). Specifically, the model takes the sequence $([s], R_1, R_2, \ldots, R_{K-1})$ as input and predicts the corresponding sequence $(R_1, R_2, R_3, \ldots, R_K)$. Training is guided by a standard cross-entropy loss.

$$\mathcal{L}_{CLIP} = -\frac{1}{B} \sum_{i=1}^{B} \left( \log \frac{\exp(s(e_i, z_i)/\tau)}{\sum_{j=1}^{B} \exp(s(e_i, z_j)/\tau)} + \log \frac{\exp(s(e_i, z_i)/\tau)}{\sum_{k=1}^{B} \exp(s(e_k, z_i)/\tau)} \right) \tag{4}$$

where $s$ denotes cosine similarity and $\tau$ is a learned temperature parameter that controls the sharpness of the distribution. This objective effectively creates a shared embedding space where EEG signals are mapped near their corresponding image representations and away from unrelated ones.

To further strengthen the alignment, we incorporate a direct regression objective following practices established in Benchetrit et al. (2023) and Li et al. (2024):

$$\mathcal{L}_{Combined} = \lambda \mathcal{L}_{CLIP} + (1 - \lambda) \mathcal{L}_{MSE} \tag{5}$$

where $\mathcal{L}_{MSE}$ is the mean squared error between EEG and image embeddings, and $\lambda$ (set to 0.8 in our experiments) balances the two objectives. This dual-objective approach provides more stable training dynamics and improves the precision of the EEG-to-image mapping by combining the structural alignment properties of contrastive learning with the point-wise precision of direct regression.

## 2.3 AUTOREGRESSIVE EEG-TO-IMAGE GENERATION

With aligned EEG representations in hand, we turn to the challenge of generating corresponding images. Rather than employing complex diffusion-based pipelines, we adopt a hierarchical autoregressive approach inspired by VAR (Tian et al., 2024). This framework enables direct, progressive image generation from EEG embeddings through a coarse-to-fine refinement process.

The architecture consists of two key components:

1) **Multi-scale image tokenization**: A pre-trained VQ-VAE tokenizes images into a hierarchy of discrete representations at multiple resolutions. Given an image $I$, the tokenizer produces a feature map $F \in \mathbb{R}^{h \times w \times d}$ that is quantized into $K$ multi-scale residual maps $(R_1, R_2, \ldots, R_K)$, where each $R_k$ has resolution $h_k \times w_k$ that progressively increases with $k$.

These residual maps can be combined to progressively reconstruct the full-resolution feature map:

$$F_k = \sum_{i=1}^{k} \text{up}(R_i, (h, w)), \tag{6}$$

where $\text{up}(\cdot)$ denotes bilinear upsampling, and $F_k$ represents the accumulated feature map after incorporating the first $k$ residuals. This formulation allows the image to be constructed incrementally, from coarse structures to fine details.

2) **Next-scale prediction transformer**: A decoder-only transformer is trained to autoregressively predict these residual maps from EEG embeddings $e$. The model learns to generate increasingly detailed visual representations according to:

$$p(R_1, \ldots, R_K) = \prod_{k=1}^{K} p(R_k \mid R_1, \ldots, R_{k-1}, e), \tag{7}$$

where the sequence $(R_1, \ldots, R_{k-1}, e)$ provides the context for predicting the next-scale residual $R_k$.

This formulation is particularly appropriate for neural decoding because it mirrors theories of hierarchical visual processing in the brain, where perception progresses from coarse features to increasingly fine details. The EEG embedding $e$ serves as the initial neural representation of the perceived image, and the transformer progressively elaborates this representation across multiple scales.

In practice, as shown in Fig. 2, the EEG embedding $e \in \mathbb{R}^d$ is first projected to the transformer's hidden dimension $h$ to create a special token $[s]$, which initiates the generation process. For each subsequent scale $k > 1$, the model processes the appropriately downsampled version of the previous cumulative feature map:

$$\widetilde{F}_{k-1} = \text{down}(F_{k-1}, (h_k, w_k)), \tag{8}$$

where $\text{down}(\cdot)$ represents bilinear downsampling to match the target resolution $(h_k, w_k)$ of the current scale.

During training, we employ a block-wise causal attention mask to ensure the model only attends to the appropriate context when predicting each scale. During inference, the process begins with the EEG embedding and autoregressively generates each scale until reaching the final resolution, at which point the multi-scale VQ-VAE decoder transforms the predicted feature map $\widetilde{F}_K$ into a complete image.

## 3 EXPERIMENTS

### 3.1 EXPERIMENTAL SETUP

We primarily evaluate our method on the THINGS-EEG dataset (Grootswagers et al., 2022), which serves as a widely adopted benchmark for EEG-based visual decoding. To further verify the versatility of AVDE, we additionally conduct experiments on the EEG-ImageNet dataset (Zhu et al., 2024), with results reported in Appendix C.

**Dataset Overview.** The THINGS-EEG dataset (Grootswagers et al., 2022) contains EEG recordings from 10 participants collected under a rapid serial visual presentation (RSVP) paradigm. The training set consists of 1,654 object concepts, each associated with 10 images presented four times, yielding a total of 66,160 EEG trials. The test set includes 200 distinct concepts, each represented by a single image repeated 80 times, resulting in 16,000 EEG trials. To mitigate habituation effects, both training and test images are presented in a pseudorandom order. Each image is displayed for 100 milliseconds, followed by a 100-millisecond blank screen to reduce blink-related and other artifacts. EEG signals were recorded from 63 channels, band-pass filtered between 0.1 Hz and 100 Hz, and sampled at 1,000 Hz.

**Data Preprocessing.** Following the practice in Song et al. (2023) and Li et al. (2024), we segment the EEG data into epochs spanning 0 to 1,000 ms relative to stimulus onset and apply baseline

Table 1: Overall accuracy of 200-way zero-shot retrieval under both within-subject and cross-subject settings. Each cell presents the Top-1 accuracy on the first line and the Top-5 accuracy on the second line. Results are averaged over five different random seeds; corresponding standard deviation values are presented in Table 12. For each subject, the highest accuracy values are indicated in bold.

| Method | Sub-01 | Sub-02 | Sub-03 | Sub-04 | Sub-05 | Sub-06 | Sub-07 | Sub-08 | Sub-09 | Sub-10 | Ave |
|---|---|---|---|---|---|---|---|---|---|---|---|
| **Within-subject**: train and test on one subject | | | | | | | | | | | |
| EEGNetV4 (Lawhern et al., 2018) | 0.144 | 0.159 | 0.202 | 0.224 | 0.132 | 0.129 | 0.198 | 0.246 | 0.184 | 0.237 | 0.186 |
| | 0.391 | 0.398 | 0.432 | 0.517 | 0.289 | 0.402 | 0.467 | 0.549 | 0.419 | 0.543 | 0.441 |
| EEGConformer (Song et al., 2022) | 0.095 | 0.108 | 0.142 | 0.155 | 0.088 | 0.081 | 0.122 | 0.164 | 0.109 | 0.151 | 0.122 |
| | 0.261 | 0.274 | 0.306 | 0.371 | 0.198 | 0.280 | 0.318 | 0.405 | 0.298 | 0.392 | 0.310 |
| NICE (Song et al., 2023) | 0.201 | 0.192 | 0.212 | 0.224 | 0.144 | 0.261 | 0.269 | 0.382 | 0.234 | 0.298 | 0.242 |
| | 0.479 | 0.369 | 0.538 | 0.504 | 0.316 | 0.563 | 0.557 | 0.674 | 0.532 | 0.586 | 0.512 |
| ATM (Li et al., 2024) | 0.232 | 0.188 | 0.273 | 0.280 | 0.168 | 0.280 | 0.268 | 0.393 | 0.245 | 0.372 | 0.269 |
| | 0.512 | 0.432 | 0.570 | 0.541 | 0.395 | 0.592 | 0.537 | **0.715** | 0.512 | 0.677 | 0.548 |
| AVDE (Ours) | **0.250** | **0.241** | **0.275** | **0.298** | **0.254** | **0.335** | **0.274** | **0.417** | **0.261** | **0.395** | **0.300** |
| | **0.552** | **0.510** | **0.586** | **0.547** | **0.503** | **0.603** | **0.552** | 0.713 | **0.521** | **0.730** | **0.582** |
| **Cross-subject**: leave one subject out for test | | | | | | | | | | | |
| EEGNetV4 | 0.086 | 0.082 | 0.073 | 0.113 | 0.092 | 0.101 | 0.056 | 0.084 | 0.074 | 0.124 | 0.089 |
| | 0.232 | 0.226 | 0.171 | 0.257 | 0.217 | 0.224 | 0.182 | 0.231 | 0.196 | 0.305 | 0.224 |
| EEGConformer | 0.069 | 0.066 | 0.058 | 0.090 | 0.074 | 0.081 | 0.045 | 0.067 | 0.059 | 0.099 | 0.071 |
| | 0.197 | 0.193 | 0.146 | 0.217 | 0.185 | 0.191 | 0.156 | 0.198 | 0.167 | 0.260 | 0.191 |
| NICE | 0.103 | 0.100 | 0.086 | 0.127 | 0.091 | 0.146 | **0.102** | 0.112 | 0.098 | 0.169 | 0.113 |
| | 0.286 | 0.257 | 0.206 | 0.323 | 0.183 | 0.341 | **0.268** | 0.239 | 0.242 | 0.386 | 0.273 |
| ATM | 0.121 | 0.128 | 0.082 | 0.127 | 0.094 | 0.107 | 0.083 | 0.122 | 0.096 | 0.171 | 0.115 |
| | 0.296 | 0.302 | **0.224** | 0.293 | 0.249 | 0.259 | 0.257 | 0.296 | 0.247 | 0.381 | 0.280 |
| AVDE (Ours) | **0.141** | **0.170** | **0.091** | **0.152** | **0.125** | **0.173** | 0.074 | **0.185** | **0.132** | **0.180** | **0.143** |
| | **0.322** | **0.384** | 0.218 | **0.325** | **0.324** | **0.386** | 0.204 | **0.401** | **0.336** | **0.393** | **0.329** |

correction using the mean signal from the 200 ms pre-stimulus interval. All electrodes are preserved, and the data are downsampled to 200 Hz. Given that EEG amplitudes typically range from –0.1 mV to 0.1 mV, we normalize the signals by scaling them with respect to 0.1 mV, resulting in values primarily distributed between –1 and 1. For the test set, EEG responses corresponding to each image are averaged across repetitions to improve the signal-to-noise ratio.

**Implementation Details.** We initialize the EEG encoder and the visual autoregressive (VAR) transformer with the pre-trained weights provided in the official GitHub repositories of LaBraM (Jiang et al., 2024) and VAR (Tian et al., 2024), respectively. The EEG encoder is trained using the AdamW optimizer with an initial learning rate of 2e-3, a weight decay of 0.05, and a minimum learning rate of 1e-5. The batch size is set to 128. For the VAR transformer, we configure the model with a depth of 16 and train it using the AdamW optimizer with $\beta_1 = 0.9$, $\beta_2 = 0.95$, a base learning rate of 2e-5, a weight decay of 0.05, a global batch size of 512, and 50 training epochs. Additional hyperparameter details are provided in Appendix B. During generation, we employ classifier-free guidance (CFG) with a ratio of 4.0 and apply top-k sampling with $k = 900$. All the experiments are conducted on Linux servers equipped with four NVIDIA A100 (40G) GPUs and Python 3.10.16 + PyTorch 2.5.1 + CUDA 12.4 environment.

**Evaluation.** We assess the effectiveness of AVDE on both image retrieval and reconstruction tasks. For the retrieval task, we compute the cosine similarity between the EEG embeddings generated by the EEG encoder and the CLIP image embeddings of 200 test concepts. Retrieval performance is evaluated based on the probability that the ground truth concept appears among the top-K candidates (K = 1 or 5). For the reconstruction task, we adopt standard evaluation metrics following prior work (Scotti et al., 2023; Li et al., 2024) to quantify the similarity between reconstructed and ground truth visual stimuli: (1) PixCorr – pixel-wise correlation; (2) SSIM – Structural Similarity Index Measure; (3) SwAV – average correlation distance computed from SwAV-ResNet50 (Caron et al., 2020) features; and (4) Two-way identification using pretrained neural networks (AlexNet (Krizhevsky et al., 2012) layers 2 and 5, Inception (Szegedy et al., 2015), and CLIP). Two-way identification is treated as a bidirectional retrieval task, as described in Ozcelik & VanRullen (2023).

## 3.2 RETRIEVAL PERFORMANCE

Table 1 presents a quantitative evaluation of EEG-based image retrieval performance, comparing our proposed method, AVDE, with several baseline approaches. Remarkably, AVDE achieves a top-1 accuracy of 0.300 and a top-5 accuracy of 0.582 in the zero-shot EEG-to-image retrieval task under the within-subject setting. Under the more challenging cross-subject setting, it attains a top-1 accuracy of 0.143 and a top-5 accuracy of 0.329. These results represent a substantial improvement over existing state-of-the-art methods, highlighting the effectiveness of our approach.

The strong performance of AVDE underscores the utility of the LaBraM-based EEG encoder, which benefits significantly from large-scale pre-training. This pre-training enables the encoder to generalize more effectively across individuals and extract semantically meaningful features from raw EEG signals. These high-quality EEG embeddings serve as a robust foundation for the subsequent visual decoding stage, thereby facilitating more accurate and coherent EEG-to-image generation. Overall, these findings demonstrate the potential of leveraging pre-trained architectures to enhance the performance of EEG-based inference systems.

## 3.3 RECONSTRUCTION PERFORMANCE

Since Subject-08 exhibits the highest retrieval performance, we follow the convention of prior works (Li et al., 2024; Zhang et al., 2025; Xiao et al., 2025) by quantitatively evaluating reconstruction performance on Subject-08 to ensure a fair comparison. As shown in Table 2, AVDE outperforms existing approaches, achieving the highest scores across both high-level and low-level metrics. These results indicate that AVDE improves not only low-level visual fidelity but also high-level semantic consistency in the reconstructed images. Additional results for other subjects are provided in Appendix E.

Table 2: Quantitative assessments of EEG-based visual reconstruction quality on Subject-08.

| Method | Low-level | | High-level | | | | |
|---|---|---|---|---|---|---|---|
| | PixCorr ↑ | SSIM ↑ | AlexNet(2) ↑ | AlexNet(5) ↑ | Inception ↑ | CLIP ↑ | SwAV ↓ |
| Li et al. (2024) | 0.160 | 0.345 | 0.776 | 0.866 | 0.734 | 0.786 | 0.582 |
| GeoCap | 0.148 | 0.380 | 0.813 | 0.877 | 0.712 | 0.791 | 0.582 |
| CognitionCapturer | 0.175 | 0.366 | 0.760 | 0.610 | 0.721 | 0.744 | 0.577 |
| AVDE (Ours) | **0.188** | **0.396** | **0.817** | **0.889** | **0.765** | **0.795** | **0.557** |

Qualitative results in Figure 3 further substantiate the quantitative evaluations. Compared to previous methods, AVDE reconstructs images that more closely resemble the ground-truth stimuli in terms of both structure and recognizable content. Whereas earlier models often yield semantically ambiguous reconstructions, our approach recovers finer details and clearer object shapes, benefiting from progressive multi-scale refinement and the informative EEG feature initialization.

## 3.4 EFFICIENCY ANALYSIS

We further analyze the inference efficiency of AVDE by comparing it against previous state-of-the-art methods. Since most existing approaches adopt a similar unCLIP pipeline and utilize the same diffusion model (SDXL (Podell et al., 2023)) for image generation, we select Li et al. (2024) as a representative due to its widespread adoption and open-source availability. To comprehensively assess both computational and spatial efficiency, we evaluate the following metrics: (1) FLOPs — the number of floating-point operations; (2) Inference time — the GPU time required for generation; and (3) Memory usage — the peak GPU memory usage during inference. All metrics are measured using PyTorch's built-in profiler on a single NVIDIA A100 GPU. The batch size is set to 1, corresponding to the resource cost of generating a single image. As summarized in Table 3, AVDE achieves faster image generation and lower memory consumption compared to prior state-of-the-art methods, demonstrating its superior suitability for practical applications.

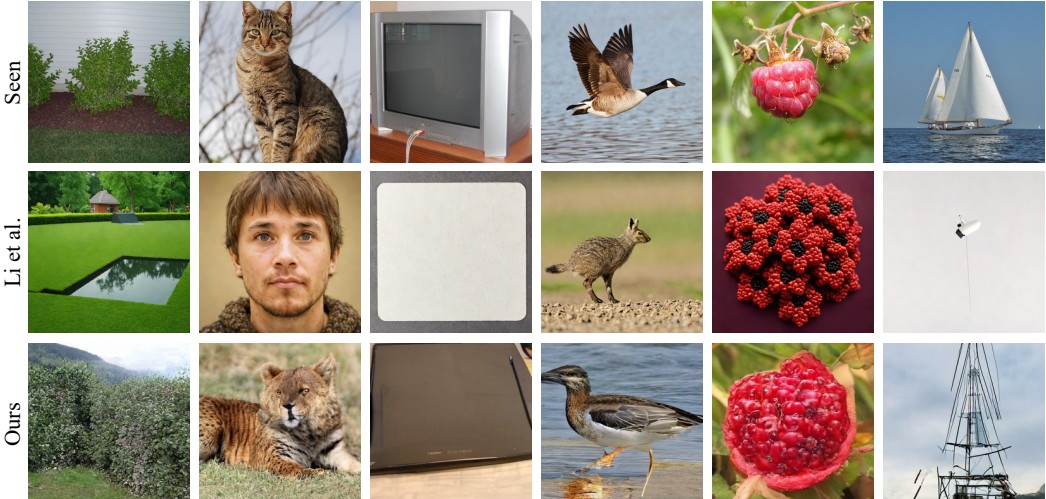

Figure 3: Qualitative Comparison of Visual Reconstruction Performance. Selected reconstruction results from subject-08 demonstrate that the visual stimuli reconstructed by our method preserve finer-grained features, suggesting improved fidelity and detail compared to alternative approaches.

Table 3: Comparison with state-of-the-art method on inference efficiency. All metrics are evaluated using PyTorch's built-in profiler on a single NVIDIA A100 GPU. The batch size is fixed at 1, and results are averaged over 200 runs to ensure stability and reliability.

| Method | Params (M) | Steps | FLOPs (G) | Inference Time (ms) | Memory Usage (MB) |
|---|---|---|---|---|---|
| Li et al. (2024) | 3818.1 | 4 | 8738.6 | 310.4 | 4826.73 |
| AVDE (Ours) | 425.3 | 10 | 1350.5 | 91.2 | 1809.63 |

## 3.5 ABLATION STUDY

To evaluate the contribution of each core component in AVDE, namely the pre-trained EEG encoder and the autoregressive generative framework, we perform the following experiments: (1) Encoder substitution (ATM/EEGNet/NICE + VAR): These experiments replace the LaBraM encoder with other widely used EEG encoders (2) Generative framework substitution (LaBraM + Li et al., unCLIP baseline): This setting replaces the VAR generative framework with a standard unCLIP pipeline (Li et al., 2024). (3) Model substitution with diffusion models (LaBraM + LDM-4 (Rombach et al., 2022) / DiT-XL (Peebles & Xie, 2023)): These experiments replace the VAR model with slightly larger diffusion models trained under the same conditions. As shown in Table 4, performance degrades when the EEG encoder is replaced, underscoring the value of high-quality embeddings from the pre-trained encoder for accurate visual reconstruction. Similarly, replacing the autoregressive framework results in a substantial drop in performance, suggesting that our overall training strategy more effectively aligns the distributional characteristics of EEG signals with those of natural images.

## 3.6 ANALYSIS OF INTERMEDIATE OUTPUTS

Given that the "next-scale prediction" strategy employed in AVDE constitutes a progressive generative process, we examine how the model incrementally extracts and interprets visual information from EEG signals throughout this procedure. To this end, we visualize all intermediate reconstructions by accumulating the feature maps at each scale and decoding them into images using the decoder of a pre-trained multi-scale VQ-VAE. Formally, the set of cumulative feature maps is denoted as $\{F_k \mid k = 1, 2, \ldots, K\}$, where $K$ represents the total number of scales, and each $F_k$ is computed as described in Equation 6.

As shown in Figure 4, the generative process in AVDE exhibits notable parallels to the hierarchical organization of human visual perception. In the early stages of generation, the model produces

Table 4: Impact of using different EEG encoders or generative framework on the reconstruction performance. The results are averaged over all subjects.

| Method | Low-level | | High-level | | | | |
|---|---|---|---|---|---|---|---|
| | PixCorr ↑ | SSIM ↑ | AlexNet(2) ↑ | AlexNet(5) ↑ | Inception ↑ | CLIP ↑ | SwAV ↓ |
| LaBraM+VAR | **0.147** | **0.366** | **0.766** | **0.835** | **0.724** | **0.747** | **0.586** |
| ATM+VAR | 0.141 | 0.351 | 0.752 | 0.821 | 0.711 | 0.731 | 0.601 |
| EEGNet+VAR | 0.132 | 0.323 | 0.733 | 0.803 | 0.687 | 0.712 | 0.627 |
| NICE+VAR | 0.136 | 0.341 | 0.742 | 0.812 | 0.701 | 0.719 | 0.613 |
| LaBraM+Li et al. (2024) | 0.138 | 0.346 | 0.746 | 0.817 | 0.707 | 0.726 | 0.606 |
| LaBraM+LDM-4 | 0.139 | 0.343 | 0.750 | 0.825 | 0.713 | 0.731 | 0.609 |
| LaBraM+DiT-XL/2 | 0.143 | 0.354 | 0.761 | 0.829 | 0.715 | 0.735 | 0.594 |

coarse features—mirroring the role of the retina and primary visual cortex (V1), which primarily encode low-level visual attributes such as edges and color gradients (Tong, 2003; Tootell et al., 1998). As the process continues, mid-level features begin to emerge, resembling the functional role of V2 and V4 in integrating contours and object-level structures (Hegdé & Van Essen, 2007). In the final stages, the model reconstructs semantically rich and coherent imagery, analogous to the activity in higher-order visual regions such as the inferotemporal cortex, where holistic object representations are formed (Tanaka, 1996).

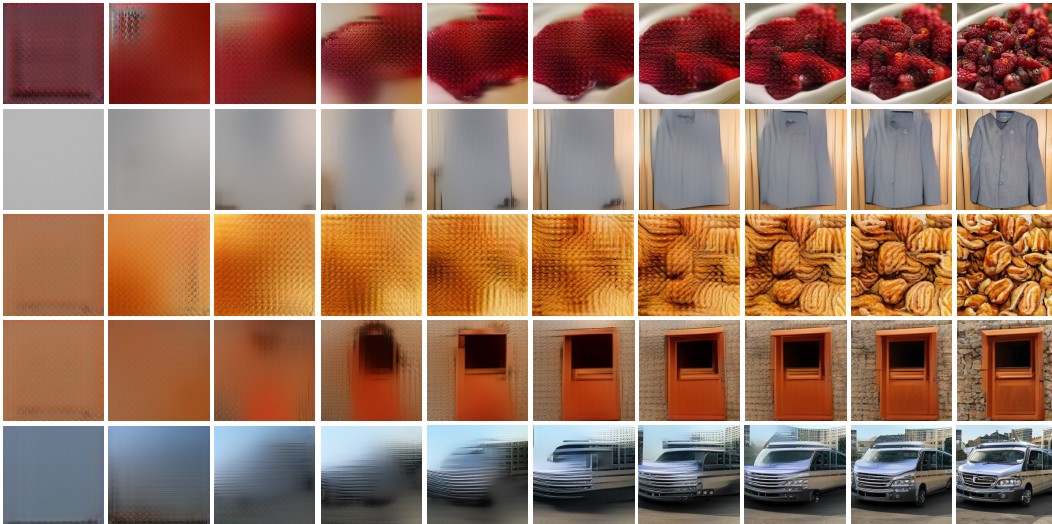

Figure 4: Intermediate reconstructions generated by AVDE across 10 progressive scales. Each row corresponds to a distinct EEG-evoked reconstruction instance, and each column represents the cumulative output up to a given scale. This process reflects the hierarchical nature of human visual perception, drawing parallels to the function of successive cortical visual areas (e.g., V1, V2/V4, and IT).

To examine the contribution of each intermediate scale to the generative process, we quantify the correlations between intermediate image features and EEG features derived from different brain regions. As shown in Figure 5a, EEG electrodes are grouped into five regions: frontal, temporal, central, parietal, and occipital. For each region, mean channel embeddings are computed using the EEG encoder, while for each intermediate scale, image embeddings are obtained from the CLIP image encoder. The cosine similarity between each region–scale pair is then calculated to assess their correspondence.

Since the generative process is cumulative, we compute the stepwise increase at each scale to capture the incremental information contributed by that scale. As shown in Figure 5c, the step increase for occipital regions peaks at early scales and gradually diminishes thereafter. In contrast, the temporal and parietal regions exhibit relatively sustained step increases across early and middle scales, fol-

lowed by a decline in later scales. The frontal and central regions, however, show low step increases initially, which progressively rise and peak at late scales. These results suggest that the intermediate scales reflect the functional roles of different brain regions during visual processing.

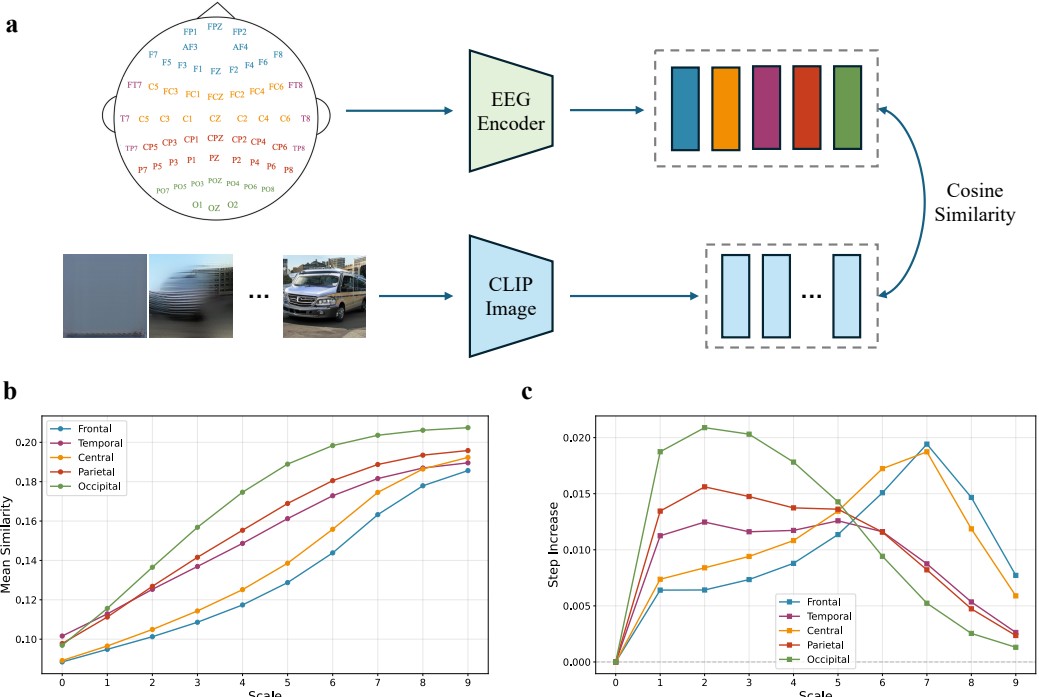

Figure 5: Analysis of similarities between intermediate scales and brain regions. (a) The mean channel embeddings from five brain regions are compared with the intermediate image embeddings. Cosine similarity is used as the measure. (b) Since the generative process is cumulative, the similarities generally increase as more scales are involved. (c) Stepwise increase captures the incremental information contributed by each scale. The step increase for occipital regions peaks at early scales and gradually diminishes thereafter. The temporal and parietal regions exhibit relatively sustained step increases across early and middle scales, followed by a decline in later scales. The frontal and central regions show low step increases initially, which progressively rise and peak at late scales.

## 4 CONCLUSION

In this work, we presented AVDE, a novel autoregressive framework for visual decoding from EEG signals that addresses key limitations of existing approaches in terms of complexity, efficiency, and performance. By leveraging pre-trained EEG representations via LaBraM and replacing multi-stage diffusion pipelines with a streamlined autoregressive process, AVDE enables accurate and coherent reconstruction of visual content from noisy EEG data. Experiments on two datasets demonstrate that AVDE outperforms state-of-the-art approaches in both retrieval and reconstruction tasks while requiring only a fraction of their computational resources, making it well-suited for practical BCI applications. Moreover, the hierarchical structure of AVDE 's generative process mirrors the structure of human visual perception, highlighting its potential as a computational tool for investigating the mechanisms of visual cognition.

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

## A  RELATED WORK

### A.1  VISUAL DECODING FROM NEURAL SIGNALS

The field of visual decoding from neural signals has evolved substantially, progressing from early pattern recognition techniques to increasingly sophisticated generative models. Foundational studies (Kay et al., 2008; Naselaris et al., 2009) established key methodologies by employing Gabor wavelet-based encoding models and Bayesian inference to identify stimuli from fMRI activity patterns. These early efforts demonstrated the feasibility of decoding visual information from brain activity, laying the groundwork for more advanced approaches. Subsequent progress was marked by the integration of deep learning, particularly through representation alignment techniques. Methods such as BrainCLIP (Liu et al., 2023) and a series of related works (Benchetrit et al., 2023; Xia et al., 2024; Takagi & Nishimoto, 2023b; Qian et al., 2023) leveraged contrastive learning frameworks inspired by CLIP (Radford et al., 2021) to align neural signals with high-level visual representations. These approaches significantly enhanced decoding accuracy, even under constraints of limited training data.

A paradigm shift occurred with the introduction of generative frameworks. Mind's Eye (Scotti et al., 2023) pioneered the application of the unCLIP approach from DALLE-2 (Ramesh et al., 2022) to fMRI-based image reconstruction, incorporating a prior diffusion model to refine neural features before generation. This multi-stage framework was subsequently adapted for EEG-based visual decoding (Li et al., 2024; Zhang et al., 2025; Xiao et al., 2025), yielding visually impressive results. However, these complex pipelines introduced significant challenges: error propagation across multiple processing stages, substantial computational requirements, and difficulty maintaining coherence between the original neural signal and the generated output. These limitations highlight the need for more direct, efficient approaches that preserve the relationship between neural activity and visual reconstruction.

### A.2  VISUAL GENERATIVE MODELS

Contemporary visual generative models primarily fall into two paradigms: diffusion models and autoregressive models, each with distinct computational characteristics particularly relevant for neural decoding applications.

Diffusion models (Ho et al., 2020; Song et al., 2020; Podell et al., 2023; Peebles & Xie, 2023; Esser et al., 2024) generate images by reversing a gradual noise-addition process, iteratively refining random noise into coherent visual content through multiple denoising steps. While these models produce high-quality images with exceptional detail, they require substantial computational resources and typically involve 25-50 sequential denoising steps, limiting their applicability in resource-constrained or real-time scenarios.

Autoregressive models (Esser et al., 2021; Yu et al., 2022; Van Den Oord et al., 2017; Razavi et al., 2019; Lee et al., 2022) offer an alternative approach by discretizing images into token sequences and predicting each token conditionally on previous ones. Traditionally, these models generated images in raster-scan order (pixel-by-pixel or patch-by-patch), which limited their ability to capture global structure and produce high-resolution outputs (Yu et al., 2022). However, recent innovations in hierarchical autoregressive modeling, particularly Visual Autoregressive modeling (VAR) (Tian et al., 2024), have transformed this landscape by introducing a coarse-to-fine generation strategy called "next-scale prediction." This approach progressively elaborates visual details across multiple resolution scales, requiring significantly fewer sampling steps while maintaining global coherence.

The efficiency advantages of VAR make it particularly well-suited for neural decoding applications, where computational constraints are significant and the hierarchical nature of the generation process aligns conceptually with theories of visual processing in the brain. AVDE builds upon this paradigm, leveraging its computational efficiency and hierarchical structure to establish a more direct mapping between EEG signals and visual representations.

## B  HYPERPARAMETER SETTINGS

Table 5: Hyperparameters for the LaBraM-based EEG encoder.

| Hyperparameters | | Values |
|---|---|---|
| | Iput channels | {1,8,8} |
| | Output channels | {8,8,8} |
| Temporal Encoder | Kernel size | {15,3,3} |
| | Stride | {8,1,1} |
| | Padding | {7,1,1} |
| Transformer encoder layers | | 12 |
| Hidden size | | 200 |
| MLP size | | 800 |
| Attention head number | | 10 |
| Batch size | | 128 |
| Peak learning rate | | 2e-3 |
| Minimal learning rate | | 1e-5 |
| Learning rate scheduler | | Cosine |
| Optimizer | | AdamW |
| Adam $\beta$ | | (0.9,0.99) |
| Weight decay | | 0.05 |
| Total epochs | | 50 |
| Warmup epochs | | 5 |
| Drop path | | 0.1 |
| Layer-wise learning rate decay | | 0.8 |
| Contrastive loss factor $\lambda$ | | 0.8 |

Table 6: Hyperparameters for the visual autoregressive transformer.

| Hyperparameters | Values |
|---|---|
| Number of layers | 16 |
| Hidden size | 1024 |
| MLP size | 4096 |
| Number of heads | 16 |
| Number of scales/steps | 10 |
| Size of scales | (1, 2, 3, 4, 5, 6, 8, 10, 13, 16) |
| Batch size | 128 |
| Peak learning rate | 2e-5 |
| Minimal learning rate | 2e-6 |
| Learning rate scheduler | Cosine |
| Optimizer | AdamW |
| Adam $\beta$ | (0.9,0.95) |
| Weight decay | 0.05 |
| Total epochs | 30 |
| Warmup epochs | 0 |
| Drop path | 0.1 |
| Normalization $\epsilon$ | 1e-6 |
| Drop rate | 0 |
| Attention drop rate | 0 |
| Condition drop rate | 0.1 |

## C    RESULTS ON EEG-IMAGENET

### C.1    DATASET DESCRIPTION

To verify the generalizability of our method, we also conducted experiments on the first 8 subjects from the EEG-ImageNet dataset (Zhu et al., 2024). The visual stimuli consisted of 80 object categories drawn from a subset of ImageNet21k. Each category contained 50 manually curated images, yielding 4,000 EEG–image pairs per subject. After randomizing the category order, the 50 images in each category were sequentially presented using the RSVP paradigm, with each image shown for 500 ms. EEG signals were recorded from 62 channels, band-pass filtered between 0.5 and 80 Hz, and sampled at 1,000 Hz. Following the protocol of the original study, we used the first 30 images of each category for training and the remaining 20 for testing.

### C.2    CLASSIFICATION

Table 7: Overall accuracy of 80-way classification on EEG-ImageNet. Each cell presents the Top-1 accuracy on the first line and the Top-5 accuracy on the second line. For each subject, the highest accuracy values are indicated in bold.

| Method | Sub-01 | Sub-02 | Sub-03 | Sub-04 | Sub-05 | Sub-06 | Sub-07 | Sub-08 |
|---|---|---|---|---|---|---|---|---|
| EEGNetV4 (Lawhern et al., 2018) | 0.297 | 0.241 | 0.206 | 0.139 | 0.177 | 0.165 | 0.152 | 0.097 |
| | 0.612 | 0.607 | 0.639 | 0.402 | 0.553 | 0.689 | 0.672 | 0.357 |
| EEGConformer (Song et al., 2022) | 0.301 | 0.263 | 0.194 | 0.141 | 0.209 | 0.173 | 0.151 | 0.102 |
| | 0.629 | 0.618 | 0.636 | 0.421 | 0.552 | 0.701 | 0.667 | 0.365 |
| NICE (Song et al., 2023) | 0.293 | 0.258 | 0.205 | 0.138 | 0.196 | 0.184 | 0.149 | 0.095 |
| | 0.621 | 0.607 | 0.640 | 0.419 | 0.561 | 0.683 | 0.671 | 0.359 |
| ATM (Li et al., 2024) | 0.306 | 0.245 | 0.213 | 0.151 | 0.205 | 0.187 | 0.146 | 0.091 |
| | 0.633 | 0.624 | 0.641 | 0.428 | 0.557 | 0.699 | 0.676 | 0.353 |
| AVDE (Ours) | 0.308 | 0.270 | 0.227 | 0.154 | 0.218 | 0.329 | 0.301 | 0.144 |
| | 0.634 | 0.628 | 0.643 | 0.435 | 0.566 | 0.703 | 0.695 | 0.388 |

### C.3    RECONSTRUCTION

Table 8: Quantitative assessments of EEG-based visual reconstruction quality on EEG-ImageNet.

| Subject | Low-level | | High-level | | | | |
|---|---|---|---|---|---|---|---|
| | PixCorr ↑ | SSIM ↑ | AlexNet(2) ↑ | AlexNet(5) ↑ | Inception ↑ | CLIP ↑ | SwAV ↓ |
| Sub-01 | 0.106 | 0.283 | 0.577 | 0.664 | 0.556 | 0.539 | 0.626 |
| Sub-02 | 0.096 | 0.277 | 0.564 | 0.643 | 0.547 | 0.527 | 0.639 |
| Sub-03 | 0.095 | 0.264 | 0.541 | 0.617 | 0.525 | 0.501 | 0.651 |
| Sub-04 | 0.080 | 0.249 | 0.505 | 0.578 | 0.480 | 0.479 | 0.677 |
| Sub-05 | 0.086 | 0.253 | 0.528 | 0.592 | 0.504 | 0.482 | 0.664 |
| Sub-06 | 0.124 | 0.318 | 0.616 | 0.701 | 0.603 | 0.568 | 0.605 |
| Sub-07 | 0.119 | 0.308 | 0.599 | 0.694 | 0.581 | 0.553 | 0.613 |
| Sub-08 | 0.077 | 0.232 | 0.480 | 0.553 | 0.461 | 0.457 | 0.690 |

Table 9: Reconstruction quality on EEG-ImageNet with baselines.

(a) Li et al. (2024)

| Subject | Low-level | | High-level | | | | |
|---|---|---|---|---|---|---|---|
| | PixCorr ↑ | SSIM ↑ | AlexNet(2) ↑ | AlexNet(5) ↑ | Inception ↑ | CLIP ↑ | SwAV ↓ |
| Sub-01 | 0.099 | 0.269 | 0.559 | 0.629 | 0.529 | 0.511 | 0.648 |
| Sub-02 | 0.092 | 0.264 | 0.543 | 0.611 | 0.532 | 0.501 | 0.659 |
| Sub-03 | 0.091 | 0.255 | 0.525 | 0.603 | 0.508 | 0.480 | 0.675 |
| Sub-04 | 0.078 | 0.236 | 0.491 | 0.563 | 0.466 | 0.460 | 0.698 |
| Sub-05 | 0.081 | 0.240 | 0.504 | 0.569 | 0.485 | 0.460 | 0.692 |
| Sub-06 | 0.119 | 0.307 | 0.589 | 0.678 | 0.584 | 0.545 | 0.628 |
| Sub-07 | 0.114 | 0.296 | 0.584 | 0.669 | 0.565 | 0.538 | 0.642 |
| Sub-08 | 0.074 | 0.224 | 0.462 | 0.529 | 0.448 | 0.435 | 0.699 |

(b) CognitionCapturer (Zhang et al., 2025)

| Subject | Low-level | | High-level | | | | |
|---|---|---|---|---|---|---|---|
| | PixCorr ↑ | SSIM ↑ | AlexNet(2) ↑ | AlexNet(5) ↑ | Inception ↑ | CLIP ↑ | SwAV ↓ |
| Sub-01 | 0.105 | 0.267 | 0.551 | 0.631 | 0.536 | 0.509 | 0.662 |
| Sub-02 | 0.098 | 0.263 | 0.540 | 0.606 | 0.539 | 0.499 | 0.646 |
| Sub-03 | 0.088 | 0.262 | 0.515 | 0.593 | 0.509 | 0.481 | 0.679 |
| Sub-04 | 0.091 | 0.243 | 0.494 | 0.570 | 0.461 | 0.456 | 0.690 |
| Sub-05 | 0.075 | 0.244 | 0.492 | 0.567 | 0.483 | 0.460 | 0.690 |
| Sub-06 | 0.113 | 0.305 | 0.601 | 0.691 | 0.584 | 0.549 | 0.616 |
| Sub-07 | 0.109 | 0.293 | 0.595 | 0.662 | 0.564 | 0.553 | 0.643 |
| Sub-08 | 0.077 | 0.213 | 0.472 | 0.532 | 0.449 | 0.430 | 0.693 |

# D  RESULTS ON THINGS-MEG

## D.1  DATASET DESCRIPTION

We also verified our method on the THINGS-MEG dataset (Hebart et al., 2023) to examine its generalizability across neural modalities. The training set consists of 1854 categories, with 12 images in each category, and the test set contains 200 categories. Each image in the dataset was displayed for 500 ms. The preprocessing pipeline includes bandpass filtering of [0.1, 40] Hz, downsampling to 200 Hz and baseline correction. Similar to THINGS-EEG, we ran the retrieval and reconstruction experiments on this dataset.

## D.2  RETRIEVAL

Table 10: Overall accuracy of 200-way zero-shot retrieval on THINGS-MEG. Each cell presents Top-1 accuracy on the left and Top-5 accuracy on the right. For each subject, the highest accuracy values are indicated in bold.

| Method | Sub-01 | Sub-02 | Sub-03 | Sub-04 |
|---|---|---|---|---|
| EEGNetV4 (Lawhern et al., 2018) | 0.118 / 0.327 | 0.136 / 0.351 | 0.171 / 0.382 | 0.192 / 0.459 |
| EEGConformer (Song et al., 2022) | 0.071 / 0.218 | 0.086 / 0.234 | 0.119 / 0.259 | 0.129 / 0.324 |
| NICE (Song et al., 2023) | 0.173 / 0.427 | 0.162 / 0.322 | 0.185 / 0.487 | 0.198 / 0.449 |
| ATM (Li et al., 2024) | 0.196 / 0.456 | 0.153 / 0.381 | 0.235 / 0.512 | 0.242 / 0.489 |
| AVDE (Ours) | 0.221 / 0.498 | 0.211 / 0.466 | 0.244 / 0.533 | 0.266 / 0.494 |

## D.3  RECONSTRUCTION

Table 11: Reconstruction quality on THINGS-MEG with baselines.

(a) Li et al. (2024)

| Subject | Low-level | | High-level | | | | |
|---|---|---|---|---|---|---|---|
| | PixCorr ↑ | SSIM ↑ | AlexNet(2) ↑ | AlexNet(5) ↑ | Inception ↑ | CLIP ↑ | SwAV ↓ |
| Sub-01 | 0.119 | 0.327 | 0.691 | 0.802 | 0.681 | 0.699 | 0.642 |
| Sub-02 | 0.094 | 0.276 | 0.661 | 0.744 | 0.627 | 0.673 | 0.664 |
| Sub-03 | 0.128 | 0.315 | 0.714 | 0.826 | 0.716 | 0.728 | 0.645 |
| Sub-04 | 0.066 | 0.281 | 0.702 | 0.754 | 0.664 | 0.661 | 0.677 |

(b) AVDE (Ours)

| Subject | Low-level | | High-level | | | | |
|---|---|---|---|---|---|---|---|
| | PixCorr ↑ | SSIM ↑ | AlexNet(2) ↑ | AlexNet(5) ↑ | Inception ↑ | CLIP ↑ | SwAV ↓ |
| Sub-01 | 0.121 | 0.329 | 0.721 | 0.804 | 0.692 | 0.699 | 0.658 |
| Sub-02 | 0.109 | 0.351 | 0.732 | 0.789 | 0.681 | 0.712 | 0.649 |
| Sub-03 | 0.152 | 0.344 | 0.740 | 0.801 | 0.668 | 0.725 | 0.654 |
| Sub-04 | 0.113 | 0.333 | 0.719 | 0.808 | 0.699 | 0.742 | 0.641 |

# E    ADDITIONAL RESULTS ON THINGS-EEG

## E.1    RETRIEVAL

Table 12: Standard deviation values of Top-1 and Top-5 accuracy on the 200-way zero-shot retrieval task. These results are complement to Table 1.

| Method | Sub-01 | Sub-02 | Sub-03 | Sub-04 | Sub-05 | Sub-06 | Sub-07 | Sub-08 | Sub-09 | Sub-10 |
|---|---|---|---|---|---|---|---|---|---|---|
| **Within-subject**: train and test on one subject | | | | | | | | | | |
| EEGNetV4 Lawhern et al. (2018) | 0.012 | 0.008 | 0.008 | 0.019 | 0.020 | 0.012 | 0.015 | 0.015 | 0.017 | 0.009 |
| | 0.045 | 0.041 | 0.025 | 0.031 | 0.010 | 0.031 | 0.035 | 0.034 | 0.026 | 0.028 |
| EEGConformer Song et al. (2022) | 0.014 | 0.015 | 0.013 | 0.020 | 0.017 | 0.010 | 0.009 | 0.011 | 0.018 | 0.019 |
| | 0.009 | 0.008 | 0.044 | 0.043 | 0.017 | 0.015 | 0.026 | 0.040 | 0.018 | 0.041 |
| NICE Song et al. (2023) | 0.010 | 0.014 | 0.012 | 0.016 | 0.012 | 0.008 | 0.015 | 0.031 | 0.010 | 0.009 |
| | 0.030 | 0.028 | 0.037 | 0.039 | 0.027 | 0.043 | 0.037 | 0.033 | 0.044 | 0.042 |
| ATM Li et al. (2024) | 0.016 | 0.006 | 0.022 | 0.015 | 0.011 | 0.027 | 0.013 | 0.022 | 0.016 | 0.020 |
| | 0.017 | 0.022 | 0.015 | 0.018 | 0.011 | 0.024 | 0.026 | 0.026 | 0.023 | 0.028 |
| AVDE (Ours) | 0.018 | 0.011 | 0.015 | 0.012 | 0.017 | 0.027 | 0.018 | 0.023 | 0.012 | 0.017 |
| | 0.020 | 0.023 | 0.016 | 0.021 | 0.019 | 0.025 | 0.018 | 0.028 | 0.019 | 0.025 |
| **Cross-subject**: leave one subject out for test | | | | | | | | | | |
| EEGNetV4 Lawhern et al. (2018) | 0.012 | 0.010 | 0.009 | 0.014 | 0.013 | 0.011 | 0.012 | 0.013 | 0.011 | 0.012 |
| | 0.025 | 0.022 | 0.021 | 0.023 | 0.020 | 0.022 | 0.024 | 0.025 | 0.021 | 0.022 |
| EEGConformer Song et al. (2022) | 0.010 | 0.009 | 0.009 | 0.013 | 0.012 | 0.009 | 0.008 | 0.008 | 0.011 | 0.011 |
| | 0.009 | 0.008 | 0.021 | 0.022 | 0.014 | 0.013 | 0.017 | 0.021 | 0.015 | 0.020 |
| NICE Song et al. (2023) | 0.009 | 0.010 | 0.009 | 0.011 | 0.010 | 0.008 | 0.011 | 0.017 | 0.008 | 0.007 |
| | 0.019 | 0.018 | 0.024 | 0.025 | 0.017 | 0.021 | 0.023 | 0.020 | 0.025 | 0.023 |
| ATM Li et al. (2024) | 0.013 | 0.008 | 0.015 | 0.011 | 0.010 | 0.018 | 0.011 | 0.015 | 0.013 | 0.014 |
| | 0.015 | 0.016 | 0.013 | 0.015 | 0.011 | 0.017 | 0.018 | 0.018 | 0.016 | 0.019 |
| AVDE (Ours) | 0.014 | 0.010 | 0.012 | 0.011 | 0.013 | 0.019 | 0.014 | 0.017 | 0.011 | 0.013 |
| | 0.016 | 0.017 | 0.013 | 0.015 | 0.014 | 0.018 | 0.015 | 0.020 | 0.015 | 0.018 |

## E.2    RECONSTRUCTION

Table 13: Quantitative assessments of EEG-based visual reconstruction quality on all subjects.

| Subject | Low-level | | High-level | | | | |
|---|---|---|---|---|---|---|---|
| | PixCorr ↑ | SSIM ↑ | AlexNet(2) ↑ | AlexNet(5) ↑ | Inception ↑ | CLIP ↑ | SwAV ↓ |
| Sub-01 | 0.138 | 0.362 | 0.750 | 0.837 | 0.725 | 0.721 | 0.609 |
| Sub-02 | 0.120 | 0.388 | 0.754 | 0.811 | 0.710 | 0.737 | 0.604 |
| Sub-03 | 0.168 | 0.379 | 0.763 | 0.827 | 0.704 | 0.746 | 0.603 |
| Sub-04 | 0.126 | 0.361 | 0.748 | 0.835 | 0.731 | 0.765 | 0.593 |
| Sub-05 | 0.145 | 0.373 | 0.743 | 0.814 | 0.691 | 0.732 | 0.586 |
| Sub-06 | 0.128 | 0.364 | 0.768 | 0.832 | 0.707 | 0.717 | 0.584 |
| Sub-07 | 0.126 | 0.351 | 0.760 | 0.827 | 0.733 | 0.731 | 0.588 |
| Sub-08 | 0.188 | 0.396 | 0.817 | 0.889 | 0.765 | 0.795 | 0.557 |
| Sub-09 | 0.156 | 0.339 | 0.753 | 0.810 | 0.713 | 0.726 | 0.600 |
| Sub-10 | 0.173 | 0.349 | 0.807 | 0.871 | 0.764 | 0.800 | 0.543 |

Table 14: Reconstruction quality on THINGS-EEG with baselines.

(a) Li et al. (2024)

| Subject | Low-level | | High-level | | | | |
|---|---|---|---|---|---|---|---|
| | PixCorr ↑ | SSIM ↑ | AlexNet(2) ↑ | AlexNet(5) ↑ | Inception ↑ | CLIP ↑ | SwAV ↓ |
| Sub-01 | 0.136 | 0.354 | 0.720 | 0.831 | 0.710 | 0.728 | 0.589 |
| Sub-02 | 0.108 | 0.301 | 0.688 | 0.775 | 0.657 | 0.705 | 0.618 |
| Sub-03 | 0.145 | 0.340 | 0.747 | 0.858 | 0.751 | 0.759 | 0.590 |
| Sub-04 | 0.075 | 0.305 | 0.729 | 0.782 | 0.694 | 0.688 | 0.623 |
| Sub-05 | 0.133 | 0.352 | 0.738 | 0.791 | 0.637 | 0.707 | 0.612 |
| Sub-06 | 0.140 | 0.363 | 0.753 | 0.834 | 0.686 | 0.742 | 0.595 |
| Sub-07 | 0.155 | 0.360 | 0.769 | 0.851 | 0.684 | 0.697 | 0.608 |
| Sub-08 | 0.163 | 0.345 | 0.786 | 0.868 | 0.730 | 0.770 | 0.575 |
| Sub-09 | 0.134 | 0.338 | 0.736 | 0.788 | 0.626 | 0.652 | 0.606 |
| Sub-10 | 0.112 | 0.329 | 0.735 | 0.843 | 0.674 | 0.708 | 0.570 |

(b) CognitionCapturer (Zhang et al., 2025)

| Subject | Low-level | | High-level | | | | |
|---|---|---|---|---|---|---|---|
| | PixCorr ↑ | SSIM ↑ | AlexNet(2) ↑ | AlexNet(5) ↑ | Inception ↑ | CLIP ↑ | SwAV ↓ |
| Sub-01 | 0.148 | 0.334 | 0.741 | 0.626 | 0.666 | 0.711 | 0.592 |
| Sub-02 | 0.147 | 0.344 | 0.764 | 0.618 | 0.661 | 0.725 | 0.590 |
| Sub-03 | 0.140 | 0.307 | 0.715 | 0.549 | 0.690 | 0.710 | 0.603 |
| Sub-04 | 0.126 | 0.355 | 0.801 | 0.660 | 0.701 | 0.765 | 0.543 |
| Sub-05 | 0.130 | 0.343 | 0.731 | 0.639 | 0.594 | 0.655 | 0.611 |
| Sub-06 | 0.122 | 0.337 | 0.748 | 0.620 | 0.646 | 0.688 | 0.630 |
| Sub-07 | 0.145 | 0.355 | 0.777 | 0.623 | 0.731 | 0.721 | 0.576 |
| Sub-08 | 0.175 | 0.366 | 0.760 | 0.610 | 0.721 | 0.744 | 0.577 |
| Sub-09 | 0.148 | 0.337 | 0.731 | 0.623 | 0.625 | 0.692 | 0.605 |
| Sub-10 | 0.152 | 0.389 | 0.773 | 0.664 | 0.657 | 0.736 | 0.569 |

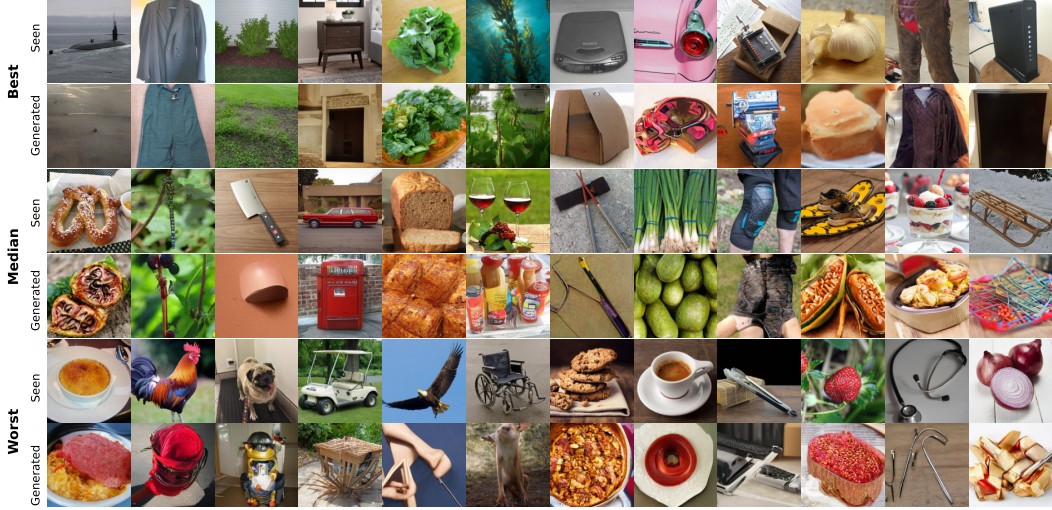

Figure 6: Subject-08's Best, Medium, and Worst reconstructions selected based on PixCorr.

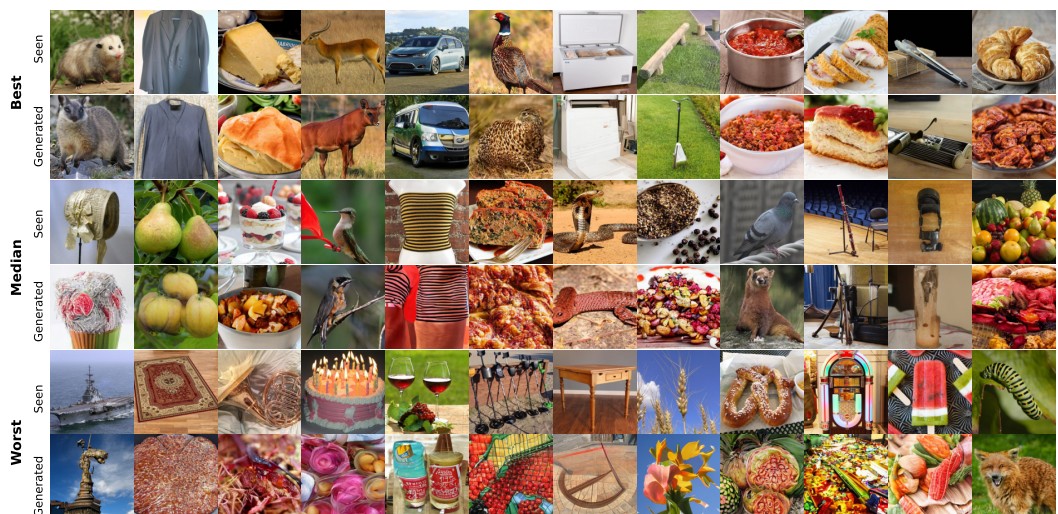

Figure 7: Subject-08's Best, Medium, and Worst reconstructions selected based on SwAV.

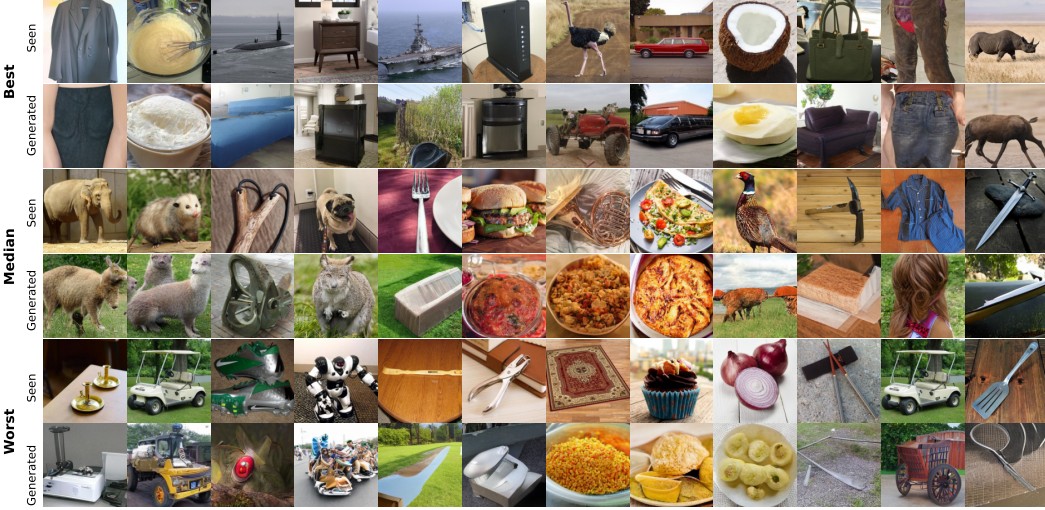

Figure 8: Subject-10's Best, Medium, and Worst reconstructions selected based on PixCorr.

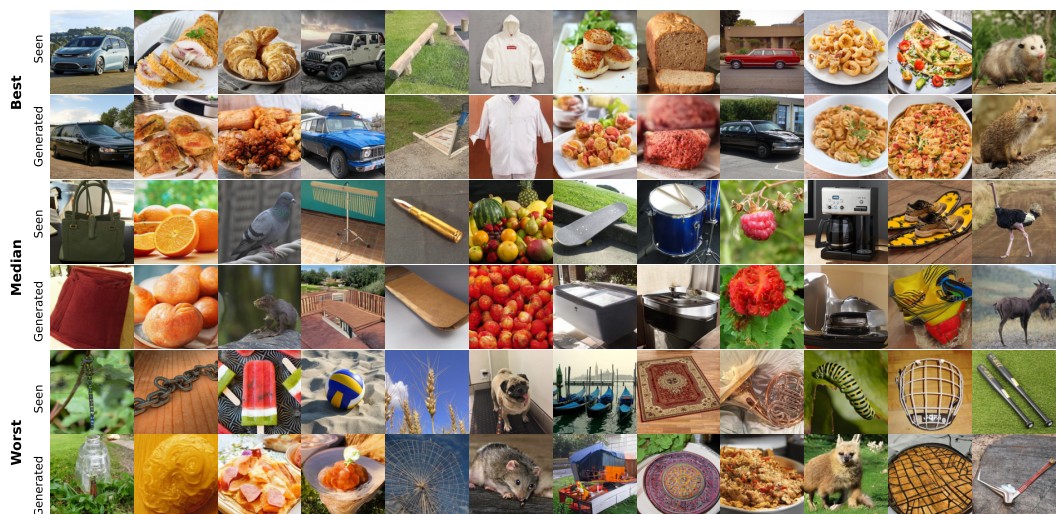

Figure 9: Subject-10's Best, Medium, and Worst reconstructions selected based on SwAV.

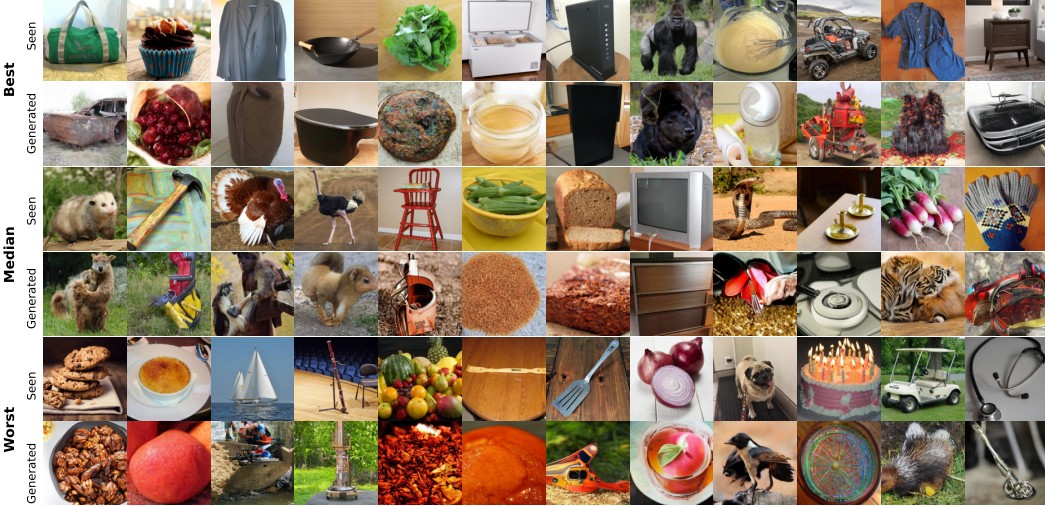

Figure 10: Subject-03's Best, Medium, and Worst reconstructions selected based on PixCorr.

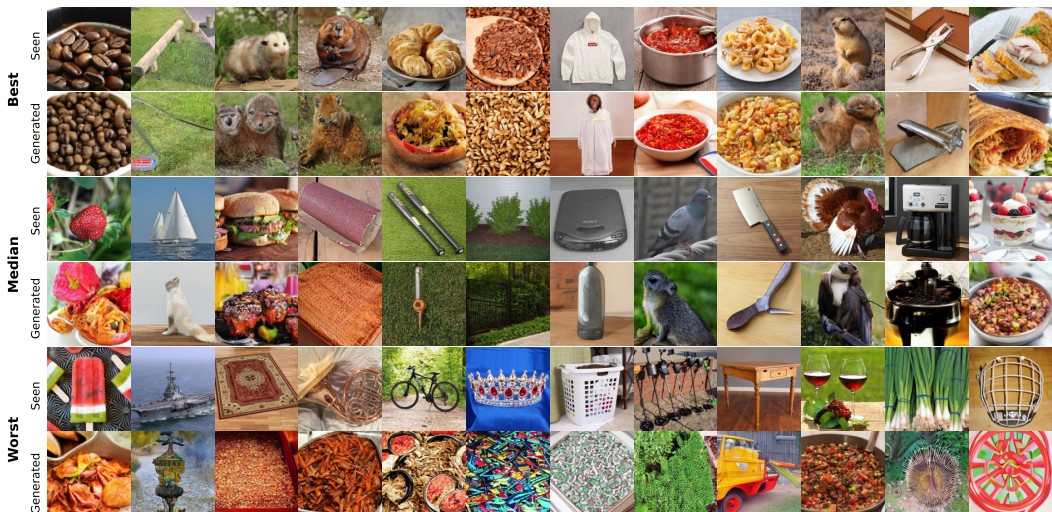

Figure 11: Subject-03's Best, Medium, and Worst reconstructions selected based on SwAV.

