# OpenReview forum: "Autoregressive Visual Decoding from EEG Signals"
_ICLR.cc/2026/Conference — ICLR 2026 Poster_

### Official Review · Reviewer_p5Kj · 2025-10-20

**Soundness:** 2
**Presentation:** 3
**Contribution:** 2
**Rating:** 4
**Confidence:** 4

**Summary:**

This paper proposes AVDE, a framework for reconstructing visual images from EEG signals, aiming to improve upon existing methods that often rely on complex, computationally expensive diffusion pipelines. AVDE uses a two-stage process: first, it fine-tunes a large pre-trained EEG model (LaBraM) using contrastive learning to align EEG and image (CLIP) representations. Second, it employs an autoregressive transformer, inspired by VAR's "next-scale prediction" strategy and using a pre-trained VQ-VAE, to generate images progressively from the aligned EEG embedding. The authors report state-of-the-art results on retrieval and reconstruction tasks using the THINGS-EEG dataset, highlighting significant improvements in efficiency (fewer parameters, faster inference) compared to diffusion-based predecessors.

**Strengths:**

Demonstrates substantial reductions in parameter count (~90%), computational cost (FLOPs), inference time, and memory usage compared to a representative diffusion-based baseline, making it more viable for practical BCI applications.
Successfully leverages large pre-trained models (LaBraM, VAR) through fine-tuning and contrastive alignment, highlighting the benefits of transfer learning in this domain.

**Weaknesses:**

The ablation (Table 4) does not cleanly isolate the contribution of the autoregressive generative model (VAR) versus a comparable diffusion model when using the same high-quality LaBraM encoder. This makes it difficult to definitively conclude that the VAR approach itself is superior.
The framework primarily combines existing powerful components (LaBraM, VAR architecture, VQ-VAE, CLIP alignment) rather than introducing fundamentally new techniques.
Linking specific generation scales to cortical areas (V1/V2/IT) is speculative without stronger supporting evidence .

**Questions:**

Could you clarify the ablation study in Table 4? To fairly assess the contribution of the VAR framework, could you provide results for a LaBraM + Diffusion baseline trained under identical conditions (using the same fine-tuned LaBraM encoder) as your main LaBraM + VAR model?

Regarding the efficiency comparison in Table 3: How does AVDE compare to the diffusion baseline if the latter is optimized for speed using fewer sampling steps (e.g., 10-20, if image quality remains acceptable)?

Table 2 focuses on Subject-08. How robust are the reconstruction performance gains across other subjects, particularly those with lower retrieval scores? An average comparison across subjects for reconstruction would be valuable.

---

> ### Author Response · Authors · 2025-11-20
> **Response to reviewr p5Kj (1/2)**
>
> Thank you for your thoughtful and constructive feedback. Please see our reponses below.
>
> > [W1, Q1] Could you clarify the ablation study in Table 4? To fairly assess the contribution of the VAR framework, could you provide results for a LaBraM + Diffusion baseline trained under identical conditions (using the same fine-tuned LaBraM encoder) as your main LaBraM + VAR model?
>
> We fully agree that including a LaBraM + Diffusion baseline trained under identical conditions would strengthen the ablation study. To address this, we have conducted additional experiments using fine-tuned LaBraM as the encoder paired with either LDM-4 (400M) or DiT-XL/2 (675M) as the diffusion model. Note that both selected diffusion models are larger than VAR because we were unable to find smaller diffusion models with publicly available checkpoints. We set the inference sampling steps to 250 to stay consistent with the original papers. As reported in Table 4 of the revised manuscript, neither diffusion baseline surpasses the VAR framework on the reconstruction task. These results highlight VAR’s advantage in achieving stronger performance with fewer parameters. We also explored the possibility of unfreezing SDXL (3.5B) and training it under the same setup, but this was not feasible due to computational resource constraints.
>
> After the revision, the experiments in the ablation study (Table 4) primarily fall into three categories:
> 1. Encoder substitution (ATM/EEGNet/NICE + VAR): These experiments replace the LaBraM encoder with other widely used EEG encoders to demonstrate that the high-quality, pre-trained LaBraM representations are important for accurate visual reconstruction.
> 2. Generative framework substitution (LaBraM + Li et al., unCLIP baseline): This setting replaces the VAR generative framework with a standard unCLIP pipeline to show that our overall training strategy more effectively aligns the distributional characteristics of EEG signals with those of natural images.
> 3. Model substitution with larger diffusion models (LaBraM + LDM-4 / DiT-XL): These experiments replace the VAR model with slightly larger diffusion models trained under the same conditions to demonstrate that, beyond the training pipeline, the VAR architecture itself contributes significantly to the overall performance.
>
> > [W2] The framework primarily combines existing powerful components (LaBraM, VAR architecture, VQ-VAE, CLIP alignment) rather than introducing fundamentally new techniques.
>
> We agree that our framework builds on established components such as LaBraM, VQ-VAE, CLIP alignment, and an autoregressive transformer. However, our goal is not to introduce an entirely new primitive, but to demonstrate that the way these components are integrated enables a capability that has been largely missing in EEG-to-image decoding: a more efficient and streamlined pipeline that outperforms heavier diffusion-based systems.
>
> As mentioned in the manuscript, prior methods typically adopt the unCLIP framework, where the large-scale diffusion generator (e.g., SDXL) is kept frozen. To bridge the gap between EEG and SDXL latent space, these methods train a separate diffusion prior model to transform the EEG embeddings into image embeddings that can be accepted by SDXL as conditions. This multi-stage process tends to produce suboptimal alignment between the two modalities.
>
> In contrast, our method trains (fine-tunes) the image generator directly on the EEG–image dataset rather than keeping it frozen. This design brings two key advantages. First, it reduces the number of stages and improves the overall efficiency. Second, it maintains a direct relationship between EEG signals and visual outputs, making the cross-modal alignment more effective. Therefore, taken together the differences in both the components models and the overall pipeline, we believe that the proposed framework is fundamentally different from previous approaches.

---

> ### Author Response · Authors · 2025-11-20
> **Response to reviewer p5Kj (2/2)**
>
> > [W3] Linking specific generation scales to cortical areas (V1/V2/IT) is speculative without stronger supporting evidence .
>
> We acknoledge that the statement was qualitative and required quantitative support. In response, we quantified the correlations between intermediate image features and EEG features derived from different brain regions. Specifically, for each region, we computed the mean channel embeddings using the EEG encoder, and for each intermediate scale, we extracted image embeddings from the CLIP image encoder. The cosine similarity between each region–scale pair was then calculated to quantify their correspondence.
>
> Given that the generative process is cumulative, we further computed the stepwise increase in similarity at each scale to capture the incremental information contributed by that scale. The results reveal distinct temporal–spatial patterns: the occipital regions show strong increases at early scales that gradually diminish; the temporal and parietal regions exhibit sustained increases across early and middle scales before declining; and the frontal and central regions display low initial increases that progressively rise, peaking at later scales.
>
> These findings suggest that the intermediate scales of the generative process reflect the functional hierarchy of brain regions involved in visual processing. Please refer to Figure 5 in the revised manuscript for a detailed illustration of the analysis and corresponding results.
>
> > [Q2] Regarding the efficiency comparison in Table 3: How does AVDE compare to the diffusion baseline if the latter is optimized for speed using fewer sampling steps (e.g., 10-20, if image quality remains acceptable)?
>
> Thanks for pointing this out. In our experiments, we observed that the metrics (FLOPs/Inference Time/Memory Usage) are dominated by the 4 sampling steps of SDXL. The diffusion prior model is relatively small compared to SDXL, so changing its number of sampling steps doesn't make a huge difference in the metrics. To avoid confusion, we have updated Table 3 in the revised manuscript to only report the number of steps for SDXL.
>
> > [Q3] Table 2 focuses on Subject-08. How robust are the reconstruction performance gains across other subjects, particularly those with lower retrieval scores? An average comparison across subjects for reconstruction would be valuable.
>
> Reconstruction results on other subjects are included in Appendix E of the manuscript. We focused on Subject-08 in Table 2 to stay consistent with the convention in prior studies and ensure fair comparison. That said, we do agree that an average comparison across subjects would be valuable, so we provide it in the table below.
>
> | Method              | PixCorr | SSIM   | AlexNet(2) | AlexNet(5) | Inception | CLIP   | SwAV   |
> | ------------------- | ------- | ------ | ---------- | ---------- | --------- | ------ | ------ |
> | AVDE            | 0.1468  | 0.3662 | 0.7663     | 0.8353     | 0.7243    | 0.7470 | 0.5859 |
> | Li et al. (2024)    | 0.1301  | 0.3387 | 0.7401     | 0.8221     | 0.6849    | 0.7156 | 0.5986 |
> | Zhang et al. (2025) | 0.1433  | 0.3467 | 0.7541     | 0.6187     | 0.6692    | 0.7147 | 0.5896 |

---

### Official Review · Reviewer_ZRBw · 2025-11-01

**Soundness:** 3
**Presentation:** 4
**Contribution:** 3
**Rating:** 6
**Confidence:** 4

**Summary:**

Summary:
This paper presents AVDE, a simple but effective framework for reconstructing visual stimuli from EEG signals. The AVDE method consists of two stages: (1) aligning EEG and image representations by fine-tuning a pre-trained EEG encoder (LaBraM) via contrastive learning, and (2) generating images using a hierarchical autoregressive transformer that predicts visual tokens from coarse to fine scales. The authors evaluate AVDE on two datasets (i.e., THINGS-EEG and EEG-ImageNet) and demonstrate superior performance in both image retrieval and reconstruction tasks compared to existing methods, while using only 10% of the parameters of prior diffusion-based models.

**Strengths:**

1. **next-scale prediction**. AVDE has an interesting idea (“next-scale prediction”, rather than traditional multi-stage diffusion models). It is conceptually novel. Unlike earlier methods that suffer from error propagation across multiple stages, AVDE generates visual content progressively, starting from coarse EEG embeddings and refining them to more detailed image representations. This approach improves coherence between EEG inputs and reconstructed images.

2.**Performance**
The proposed method demonstrates significant improvements in both image retrieval and reconstruction tasks, outperforming state-of-the-art methods while using only 10% of the parameters required by previous diffusion models. This is clearly evidenced by the performance comparison in Table 1 (Page 6), where AVDE shows superior accuracy in retrieval tasks (Top-1 accuracy of 0.300 and Top-5 accuracy of 0.582) under the within-subject setting, and substantial improvements in the cross-subject setting. These results illustrate the model's robustness and the efficiency of the framework.

Additionally, the reconstruction quality shown in Table 2 and Figure 3 further supports AVDE's high quality. AVDE achieves better low-level (PixCorr) and high-level (SSIM, AlexNet, Inception) metrics compared to existing methods, confirming that the method not only reconstructs clearer and more faithful images but also captures high-level semantic consistency.

3.The paper is generally well-written.

**Weaknesses:**

1. The paper’s strongest results are within-subject; cross-subject transfer remains limited and the manuscript offers little analysis of what factors drive between-subject variance. Minor

2. The efficiency comparison fixes diffusion at 50+4 steps and specific CFG/top-k settings while the proposed AR approach uses 10 steps; modern diffusion samplers (e.g., DDIM, DPM-Solver/DPMS++) can operate at 10–20 steps with competitive quality, and lighter priors are possible. As written, the comparison risks conflating algorithmic gains with configuration choices. Moderate

3. Current ablations do not isolate which components most drive gains, making it hard to attribute improvements. More ablations are needed. Moderate

4. Code cannot be accessed. Major

5. The manuscript states that “the generative process reflects the hierarchical nature of human visual perception,” but current evidence is largely qualitative. Visualization (Fig.4) reflects a coarse-to-fine spatial refinement induced by the multi-scale architecture. However, the text interprets this as evidence for a biological temporal/area hierarchy (retina→V1→V2/V4→IT). That mapping is speculative here: you do not manipulate time or measure area-selective correspondences. As written, it risks conflating spatial scale with neural hierarchy. Moderate

**Questions:**

1.**Performance on Cross-Subject and Real-World Data.**
While the paper presents strong within-subject results, the cross-subject performance (Top-1 accuracy of 0.143 and Top-5 accuracy of 0.329 in Table 1) suggests that the model's ability to generalize across different individuals is still limited. I understand that subject variability is a well-known challenge in EEG-based decoding. But including analysis of subject-specific factors (e.g., signal quality, attention level) and exploring personalization or domain adaptation strategies would further improve the quality of this study.

2.**No Real-Time or Latency Evaluation.**

The paper emphasizes efficiency but does not provide concrete metrics such as inference time or latency on standard hardware. This is particularly important for BCI applications, where real-time feedback is often critical.
In comparison, Scotti et al. (2023) and Zhang et al. (2025) highlight the computational bottlenecks when training large models on complex EEG-image tasks. This paper would benefit from a more detailed ablation study exploring how different training parameters (e.g., learning rate, batch size, number of epochs) affect resource consumption and performance.

3.**Model Interpretability**

The hierarchical, coarse-to-fine image generation process is a black-box operation, and while the paper visualizes the intermediate stages (Figure 4), it remains unclear how the model's decision-making process works in finer detail. For example, how does the EEG embedding influence specific features of the generated image at each scale? Are there any cases where the EEG signals lead to inconsistent or inaccurate image generation, and if so, how could the model be improved to handle such cases?

The lack of transparency in how the model generates certain image details from EEG signals is an issue, as Li et al. (2024) showed the brain regions and time for image reconstruction. Neural decoding methods should provide not only accurate predictions but also clear insights into how decisions are made from complex neural data. Future work could focus on improving the interpretability of the autoregressive process, possibly by integrating attention mechanisms or using explainable AI methods.


Ref:

Scotti, A., Banerjee, A., Goode, J., Shabalin, S., Nguyen, A., Dempster, A., Verlinde, N., Yundler, E., Weisberg, D., Norman, K., et al. (2023). Reconstructing the mind's eye: fMRI-to-image with contrastive learning and diffusion priors. NeurIPS

Zhang, K., He, L., Jiang, X., Lu, W., Wang, D., & Gao, X. (2025). CognitionCapturer: Decoding visual stimuli from human EEG signal with multimodal information. AAAI

Li, D., Wei, C., Li, S., Zou, J., Qin, H., & Liu, Q. (2024). Visual decoding and reconstruction via EEG embeddings with guided diffusion. NeurIPS

---

> ### Author Response · Authors · 2025-11-20
> **Response to reviewer ZRBw (1/2)**
>
> Thank you for your thoughtful and constructive feedback. Please see our reponses below.
>
> > [W1, Q1] The paper’s strongest results are within-subject; cross-subject transfer remains limited and the manuscript offers little analysis of what factors drive between-subject variance.
>
> While we acknowledge that cross-subject performance of our method remains limited, we would like to emphasize that by incorporating EEG pretraining, we have already achieved significant improvements over prior methods. As shown in Table 1 of the manuscript, AVDE achieves a 0.143 average top-1 accuracy, which represents a 24.3% relative improvement over ATM, while the relative improvement under within-subject setting is 11.5% (0.269->0.300). These results demonstrate that pretraining effectively enhances EEG decoding and represents a promising step toward mitigating between-subject variability. Looking forward, we anticipate that further scaling of data and model size, along with advances in pretraining paradigms, will continue to improve the generalizability of EEG decoding methods and reduce the impact of between-subject variance.
>
> > [W2] The efficiency comparison fixes diffusion at 50+4 steps and specific CFG/top-k settings while the proposed AR approach uses 10 steps
>
> Thanks for pointing this out. In our experiments, we observed that the metrics (FLOPs/Inference Time/Memory Usage) are dominated by the 4 sampling steps of SDXL. The diffusion prior model is relatively small compared to SDXL, so changing its number of sampling steps doesn't make a huge difference in the metrics. To avoid confusion, we have updated Table 3 in the revised manuscript to only report the number of steps for SDXL.
>
> > [W3] Current ablations do not isolate which components most drive gains, making it hard to attribute improvements. More ablations are needed.
>
> Thanks for your suggestion about the ablation study. We conducted additional experiments using fine-tuned LaBraM as the encoder paired with either LDM-4 (400M) or DiT-XL/2 (675M) as the diffusion model. Note that both selected diffusion models are larger than VAR because we were unable to find smaller diffusion models with publicly available checkpoints. We set the inference sampling steps to 250 to stay consistent with the original papers. As reported in Table 4 of the revised manuscript, neither diffusion baseline surpasses the VAR framework on the reconstruction task. These results highlight VAR’s advantage in achieving stronger performance with fewer parameters.
>
> After the revision, the experiments in the ablation study (Table 4) primarily fall into three categories:
> 1. Encoder substitution (ATM/EEGNet/NICE + VAR): These experiments replace the LaBraM encoder with other widely used EEG encoders to demonstrate that the high-quality, pre-trained LaBraM representations are important for accurate visual reconstruction.
> 2. Generative framework substitution (LaBraM + Li et al., unCLIP baseline): This setting replaces the VAR generative framework with a standard unCLIP pipeline to show that our overall training strategy more effectively aligns the distributional characteristics of EEG signals with those of natural images.
> 3. Model substitution with larger diffusion models (LaBraM + LDM-4 / DiT-XL): These experiments replace the VAR model with slightly larger diffusion models trained under the same conditions to demonstrate that, beyond the training pipeline, the VAR architecture itself contributes significantly to the overall performance.
>
> > [W4] Code cannot be accessed.
>
> The anonymous GitHub link had expired, and we have refreshed it. We apologize for the inconvenience.

---

> ### Author Response · Authors · 2025-11-20
> **Response to reviewer ZRBw (2/2)**
>
> > [W5, Q3] Model Interpretability
>
> We acknowledge that the statement was qualitative and required quantitative support. In response, we quantified the correlations between intermediate image features and EEG features derived from different brain regions. Specifically, for each region, we computed the mean channel embeddings using the EEG encoder, and for each intermediate scale, we extracted image embeddings from the CLIP image encoder. The cosine similarity between each region–scale pair was then calculated to quantify their correspondence.
>
> Given that the generative process is cumulative, we further computed the stepwise increase in similarity at each scale to capture the incremental information contributed by that scale. The results reveal distinct temporal–spatial patterns: the occipital regions show strong increases at early scales that gradually diminish; the temporal and parietal regions exhibit sustained increases across early and middle scales before declining; and the frontal and central regions display low initial increases that progressively rise, peaking at later scales.
>
> These findings suggest that the intermediate scales of the generative process reflect the functional hierarchy of brain regions involved in visual processing. Please refer to Figure 5 in the revised manuscript for a detailed illustration of the analysis and corresponding results.
>
> > [Q2] No Real-Time or Latency Evaluation.
>
> Efficiency evaluation is presented in Table 3 of the manuscript, where we report FLOPs, inference time and memory usage of the proposed method on standard hardware. All metrics are measured using PyTorch's built-in profiler on a single NVIDIA A100 GPU. The batch size is set to 1, corresponding to the resource cost of generating a single image.

---

> ### Public Comment · ~Jinho_Kim5 · 2025-11-26
> **Questions regarding reproducibility and implementation discrepancy.**
>
> First of all, thank you for this interesting work and providing the code.
>
> I am currently attempting to reproduce the image generation results (specifically for sub-08 and average of all subjects) using the provided code, but the metrics look different from those reported in the paper. I would appreciate some clarification on the implementation details to ensure I am matching your experimental setup correctly.
>
>
> **Q1. Training setup:** I have set up my environment as follows. Could you confirm if these match your internal settings?
> 1. **Image feature extraction:** OpenClip ViT-H-14 image features sourced from Li et al. (2024)'s repository (https://huggingface.co/datasets/LidongYang/EEG_Image_decode/tree/main).
> 2. **Image set:** 500 x 500 resolution images from the OSF repository (https://osf.io/y63gw/overview).
> 3. **Normalization:** I did not normalize eeg and image features when extracting them.
> 4. **Attempts:** I experimented with normalizing eeg and image features for MSE loss (and for both MSE and CLIP loss), but neither approach yielded the reported results.
>
>
> **Comparison of Results (Sub-08):**
> | Metric | Reported in Paper | Result from Provided Code (Unmodified) | Normalization for MSE |
> | :--- | :---: | :---: | :--: |
> | PixCorr | 0.188 |  0.173 | 0.177 |
> | SSIM | 0.396 | 0.425 | 0.423 |
> | AlexNet (2) | 0.817 | 0.756 | 0.766 |
> | AlexNet (5) | 0.889 | 0.857 | 0.854 |
> | Inception | 0.765 | 0.752 | 0.741 |
> | CLIP | 0.795 | 0.768 | 0.751 |
> | SwAV | 0.557 | 0.548 | 0.556 |
>
> &nbsp;
>
> **Q2. Discrepancies between paper and code:** I noticed a conflict between the text and the codebase.
> - **Normalization of embeddings:** the paper (lines 192-193) mentions "mean squared error between normalized EEG and image embeddings", but I could not find such normalization operation in the code. In addition, normalizing those two embeddings for MSE loss did not improve the performance overall.
>
> Could you clarify if the code in the repository reflects the final methods used for the paper's image generation metric tables, or if there is something I might have missed?
>
> Thank you for your time and response in advance.
>
> &nbsp;
>
> Ref:
>
> Li, D., Wei, C., Li, S., Zou, J., Qin, H., & Liu, Q. (2024). Visual decoding and reconstruction via EEG embeddings with guided diffusion. NeurIPS

---

> > ### Author Response · Authors · 2025-11-28
> >
> > Thanks for your interest our work, please see the response below.
> >
> > 1. **Normalization of embeddings**: This appears to be a typo in the manuscript and we will correct it. Thanks for pointing it out.
> > 2. **Training setup**: The code currently trains for 30 epochs and uses `vL_mean` to determine whether to save the current checkpoint. However, we later realized that `vL_mean` does not accurately reflect reconstruction quality. We therefore removed it and extended training to 100 epochs. We have uploaded a screenshot of the wandb training log (pixcorr.png) to the anonymous GitHub repository, showing how the performance metric evolves during training. While there are some fluctuations, the reconstruction quality generally improves as training progresses.
> > 3. **Hardware and env setup**: Could you confirm that you are using the same GPUs and software environment setups, as they can have significant impact on the results?

---

> > > ### Public Comment · ~Jinho_Kim5 · 2025-11-29
> > >
> > > Thank you for your response. I am continuing my reproduction efforts and have a few specific questions regarding the training dynamics and code logic:
> > >
> > >
> > > 1. **Hardware and env setup**: I'm using 2x A100 (80G) GPUs on a Linux server and the same software environment setups listed in the code and paper (tested with both python 3.10.16 and 3.10.19). I assume this hardware configuration is sufficient to reproduce the paper's results?
> > >
> > > 2. **Training setup**: Could you confirm if the default configuration in the anonymous GitHub repository is intended to reproduce the exact generation results reported in the paper? I have followed the instructions closely but am seeing discrepancies.
> > >
> > > 3. **Checkpoint saving metric**: Regarding ```vL_mean```, I also observed the same pattern that the steps with low ```vL_mean``` do not correspond to the high ```PixCorr```. As training progresses, ```vL_mean``` trends upward (I assume this is suggesting overfitting), while ```PixCorr``` and ```SSIM``` fluctuates significantly.
> > > - Does training for longer epochs (e.g., 100~) yield better high-level generation metrics despite the increasing validation loss?
> > > - Which specific metric did you use to select the checkpoint used for the paper's results?
> > >
> > > 4. **Potential logic error in code**: I believe I identified an issue in the checkpoint saving logic in ```train_eeg.py``` (Line 205). The code currently reads: ```best_updated = best_val_loss_mean < val_loss_mean```, not ```best_val_loss_mean > val_loss_mean```. Could you confirm if this logic is inverted?

---

> > > > ### Author Response · Authors · 2025-12-03
> > > >
> > > > Sorry for the late reply. OpenReview doesn’t seem to send email notifications for public comments.
> > > >
> > > > > Does training for longer epochs (e.g., 100~) yield better high-level generation metrics despite the increasing validation loss?
> > > >
> > > > Yes, training for more epochs does yield better high-level generation metrics. Please refer to alexnet2.png in the anonymous GitHub repository for the corresponding wandb training log. Prior work [1, 2] has discussed the mismatch between likelihood-based validation loss and sample quality, so this behavior seems common in generative models.
> > > >
> > > > > Which specific metric did you use to select the checkpoint used for the paper's results?
> > > >
> > > > We used `vacc_tail` to select the checkpoint, as we found that it aligns more closely with the reconstruction quality metrics.
> > > >
> > > > > Potential logic error in checkpoint saving.
> > > >
> > > > Yes, the logic here is indeed inverted. The current implementation follows the convention for accuracy-based metrics, and we did not update it when switching to loss-based metrics. However, this inversion did not significantly affect our results, since reconstruction quality continued to improve even as validation loss increased, as you already observed.
> > > >
> > > > > Could you confirm if the default configuration is intended to reproduce the results?
> > > >
> > > > Yes, the default configuration is intended to reproduce the reported results. We recommend training for additional epochs and trying different checkpoint selection metrics. We apologize for the inconvenience, and we would be happy to assist further after the review process if issues persist.
> > > >
> > > > References
> > > >
> > > > [1] Kumar, Dibyanshu, Philipp Väth, and Magda Gregorová. "Loss functions in diffusion models: A comparative study." Joint European Conference on Machine Learning and Knowledge Discovery in Databases. Cham: Springer Nature Switzerland, 2025.
> > > >
> > > > [2] Esfandiari, Yasin, et al. "Breaking the Likelihood-Quality Trade-off in Diffusion Models by Merging Pretrained Experts." arXiv preprint arXiv:2511.19434 (2025).

---

### Official Review · Reviewer_TTYP · 2025-11-01

**Soundness:** 3
**Presentation:** 3
**Contribution:** 2
**Rating:** 4
**Confidence:** 4

**Summary:**

This paper introduces AVDE, an autoregressive framework for reconstructing visual stimuli from EEG signals. The approach combines a pre-trained EEG encoder (LaBraM), fine-tuned via contrastive learning with CLIP embeddings, and a transformer-based autoregressive generator (VAR) built on a pre-trained VQ-VAE. The authors aim to simplify the complex multi-stage diffusion-based pipelines used in previous EEG-to-image decoding work. Experiments on THINGS-EEG report modest improvements in retrieval and reconstruction metrics, while significantly reducing computational cost and model size.

**Strengths:**

1.	The combination of a large-scale pre-trained EEG encoder (LaBraM) and an autoregressive generative model (VAR) is a technically reasonable and promising direction. Pretraining-based decoding is becoming a general paradigm, and applying it here may help bridge the data scarcity of EEG.
2.	VAR offers computational advantages over diffusion-based pipelines, including lower latency and reduced parameter count. The paper quantifies these gains and reports consistent, if moderate, improvements in retrieval and reconstruction tasks.
3.	The writing is clear and well-organized. Figures effectively illustrate the hierarchical decoding process, and ablations on efficiency and encoder replacement provide supporting evidence for the engineering claims.

**Weaknesses:**

1.	The research motivation remains underdeveloped. While AVDE is presented as an alternative to diffusion models, it does not fully resolve the fundamental challenges of EEG-to-image decoding. The work should be viewed as an exploratory step rather than a definitive advance.
2.	The methodological innovation is limited. The approach primarily substitutes diffusion with an autoregressive transformer and incorporates LaBraM pretraining. These are incremental modifications rather than new modeling ideas.
3.	Experimental data are restricted to THINGS-EEG, which is relatively small and homogeneous. Validation on richer or multimodal datasets (e.g., MEG or fMRI) would significantly strengthen the generality claim.
4.	The paper lacks neuroscientific or interpretability analysis. The authors claim that AVDE “reflects the hierarchical nature of human visual perception,” but no neural or representational evidence supports this statement. Analyses of temporal, spatial, or feature-level EEG correlations would be necessary to substantiate such claims.
5.	Figure 1 depicts the prior unCLIP pipeline rather than the proposed AVDE model, which may confuse readers. Showing the AVDE architecture first would better emphasize the paper’s own contribution.

**Questions:**

1.	Could you clarify the specific contribution of LaBraM—does its advantage come mainly from large-scale pretraining, or from its architectural design? Supplementary experiments isolating these factors would help clarify its role.
2.	What practical or theoretical advantages does the VAR-based autoregressive generation offer compared to diffusion models, beyond computational efficiency? Do you observe qualitative differences in reconstruction behavior?
3.	Have you considered extending the validation to other neural modalities (e.g., MEG, fMRI) or tasks to test whether AVDE generalizes beyond the THINGS-EEG dataset?

---

> ### Author Response · Authors · 2025-11-20
> **Response to reviewer TTYP (1/2)**
>
> Thank you for your thoughtful and constructive feedback. Please see our reponses below.
>
> > [W1] The research motivation remains underdeveloped.
>
> The research motivation behind this work centers on efficiency, which is an important topic in BCI development. Efficient algorithms reduce latency, power consumption, and computational load, which are essential for real-time use. However, previous studies have largely overlooked efficiency issues as they mostly focus on stacking larger models or adding increasingly complex piplines to improve on performance metrics.
>
> In contrast, our study takes a subtractive rather than additive approach. By leveraging LaBraM pretraining and replacing complex adaptation stages with a streamlined autoregressive pipeline, we significantly reduce model size as well as memory and computational overhead. Notably, these efficiency gains are achieved while also delivering improved benchmark performance. Such improvements make the proposed decoding system far more practical for real-world deployment.
>
> While we acknowledge that the proposed framework does not resolve all fundamental challenges in EEG-to-image decoding, we believe it represents a meaningful step forward. The efficiency-driven design brings this class of decoding systems closer to viable, real-world applications.
>
> > [W2] The methodological innovation is limited.
>
> We would like to clarify that, beyond substituting the diffusion model with an autoregressive transformer and incorporating LaBraM pretraining, our contribution also lies in the overall pipeline design, specifically in how EEG signals and images are aligned. As mentioned in the manuscript, prior methods typically adopt the unCLIP framework, where the large-scale diffusion generator (e.g., SDXL) is kept frozen. To bridge the gap between EEG and SDXL latent space, these methods train a separate diffusion prior model to transform the EEG embeddings into image embeddings that can be accepted by SDXL as conditions. This multi-stage process is inefficient and tends to produce suboptimal alignment between the two modalities.
>
> In contrast, our method trains (fine-tunes) the image generator directly on the EEG–image dataset rather than keeping it frozen. This design brings two key advantages. First, it reduces the number of stages and improves the overall efficiency. Second, it maintains a direct relationship between EEG signals and visual outputs, making the cross-modal alignment more effective. Therefore, taken together the differences in both the components models and the overall pipeline, we believe that the proposed framework is fundamentally different from the unCLIP-based approaches.
>
> > [W3, Q3] Experimental data are restricted to THINGS-EEG, which is relatively small and homogeneous.
>
> In addition to THINGS-EEG, we did also include experiments on EEG-ImageNet, another EEG-image dataset, with the corresponding results presented in Appendix C of the manuscript. That said, we fully agree that validation on other neural modalities would strengthen our claims, so we have conducted additional experiments on the THINGS-MEG dataset and reported the results in Appendix D of the revised manuscript.  Notably, although the LaBraM pretraining was primarily performed on EEG data, it also improved the model’s performance on MEG data. This suggests that LaBraM pretraining provides robust and transferable weight initializations that generalize across different neural modalities.
>
> > [W4] The paper lacks neuroscientific or interpretability analysis.
>
> We sincerely appreciate your suggestion to analyze temporal, spatial, and feature-level EEG correlations. In response, we quantified the correlations between intermediate image features and EEG features derived from different brain regions. Specifically, for each region, we computed the mean channel embeddings using the EEG encoder, and for each intermediate scale, we extracted image embeddings from the CLIP image encoder. The cosine similarity between each region–scale pair was then calculated to quantify their correspondence.
>
> Given that the generative process is cumulative, we further computed the stepwise increase in similarity at each scale to capture the incremental information contributed by that scale. The results reveal distinct temporal–spatial patterns: the occipital regions show strong increases at early scales that gradually diminish; the temporal and parietal regions exhibit sustained increases across early and middle scales before declining; and the frontal and central regions display low initial increases that progressively rise, peaking at later scales.
>
> These findings suggest that the intermediate scales of the generative process reflect the functional hierarchy of brain regions involved in visual processing. Please refer to Figure 5 in the revised manuscript for a detailed illustration of the analysis and corresponding results.

---

> ### Author Response · Authors · 2025-11-20
> **Response to reviewr TTYP (2/2)**
>
> > [W5] Figure 1 depicts the prior unCLIP pipeline rather than the proposed AVDE model, which may confuse readers.
>
> Thank you for pointing this out. We understand that presenting the prior unCLIP pipeline in Figure 1 may cause confusion, as it does not directly illustrate the proposed AVDE model. Our original intention was to provide context by first outlining the baseline unCLIP framework to highlight the limitations that our method aims to address.
>
> To improve clarity, we have revised the manuscript by updating the order of the two figures. Figure 1 now shows the proposed AVDE pipeline, while Figure 2 shows the prior unCLIP pipeline.
>
> > [Q1] Could you clarify the specific contribution of LaBraM—does its advantage come mainly from large-scale pretraining, or from its architectural design?
>
> We have conducted experiments on the retrieval task to see how LaBraM performs when it's trained from scratch, with results reported in the table below. These results demonstrate that LaBraM's advantage primarily comes from pretraining, while its architecture contributes to a lesser extent.
>
> | Method                    |  Sub-01   |  Sub-02   |  Sub-03   |  Sub-04   |  Sub-05   |  Sub-06   |  Sub-07   |  Sub-08   |  Sub-09   |  Sub-10   |  Ave  |
> | :------------------------ | :-------: | :-------: | :-------: | :-------: | :-------: | :-------: | :-------: | :-------: | :-------: | :-------: | :-------: |
> | LaBraM (scratch)             |   0.238   |   0.205   |   0.274   |   0.283   |   0.190   |   0.295   |   0.269   |   0.400   |   0.248   |   0.379   |   0.278   |
> |                           |   0.520   |   0.450   |   0.574   |   0.543   |   0.410   |   0.595   |   0.540   |   0.712   |   0.514   |   0.685   |   0.554   |
> | LaBraM (pretrained)                      |   0.250   |   0.241   |   0.275   |   0.298   |   0.254   |   0.335   |   0.274   |   0.417   |   0.261   |   0.395   |   0.300   |
> |                           |   0.552   |   0.510   |   0.586   |   0.547   |   0.503   |   0.603   |   0.552   |   0.713   |   0.521   |   0.730   |   0.582   |
> ****
>
>
> > [Q2] What practical or theoretical advantages does the VAR-based autoregressive generation offer compared to diffusion models, beyond computational efficiency? Do you observe qualitative differences in reconstruction behavior?
>
> A critical challenge in EEG-based image reconstruction is that current methods tend to "hallucinate" content when the neural signal is weak or absent. Because the deep generative models are kept frozen, they rely heavily on learned image priors rather than faithfully visualizing features present in the EEG signals. As a result, the models tend to fill in missing details to produce visually plausible outputs that may not accurately reflect the true stimulus. In other words, these reconstructions prioritize realism over fidelity, which can be problematic when the generated images mislead users by appearing convincing but incorrect.
>
> While AVDE is not specifically designed to resolve this issue, we do observe that it tends to produce outputs that are less realistic but better resemble true stimuli when the neural signals are weak. This behavior is advantageous in practice because it implicitly conveys uncertainty to the user, reducing the risk of misplaced trust in inaccurate reconstructions. That said, this advantage does not necessarily come from the VAR architecture, but more likely comes from the training pipeline. As mentioned in some of our earlier responses, we train (fine-tune) the image generator directly on EEG-image pairs rather than keeping it frozen. This maintains a direct relationship between EEG signals and visual outputs, making the model less likely to "hallucinate" contents. The advantages of the VAR architecture primarily lie in its efficiency and strong performance despite its small size. In principle, the same training pipeline could be applied to a larger diffusion model (e.g., SDXL), but doing so would be prohibitively slow and expensive.

---

### Official Review · Reviewer_xLwT · 2025-11-02

**Soundness:** 3
**Presentation:** 3
**Contribution:** 2
**Rating:** 6
**Confidence:** 4

**Summary:**

This paper proposes AVDE, a lightweight and efficient framework designed for decoding and reconstructing visual images from EEG signals. The authors critique current methods that depend on complex, multi-stage adaptations or computationally intensive diffusion models, arguing that these approaches are impractical for real-world applications in BCI.

**Strengths:**

+ This work thoughtfully addresses the challenge of computational overhead seen in models like diffusion, achieving remarkable results with just 10% of the parameters.

+ Moreover, the autoregressive approach shines with its simplicity and efficiency. By utilizing the EEG embedding as the initial token for the transformer, it establishes a clever architectural foundation.

+ This model outperforms more complex multi-stage and diffusion-based counterparts in both retrieval and reconstruction tasks, proving that a lighter model can indeed be more effective.

**Weaknesses:**

- Autoregressive models can sometimes face the challenge of error accumulation, where an initial mistake in an early token or pixel patch can amplify throughout the rest of the generation. State how we can address this.

- While this approach offers great efficiency and performs well on standard metrics, we may notice that the qualitative fidelity of its reconstructions doesn’t always match that of larger, slower models, such as diffusion, in more complex scenes. It's all about striking a balance, and share your thoughts on this.

- Moreover, the effectiveness of the reconstruction greatly hinges on the pre-trained image tokenizer, which translates images into tokens. If the tokenizer’s performance is lacking, it can limit the model's potential. Your insights would be useful.

**Questions:**

Please see weaknesses.

**Details Of Ethics Concerns:**

nil

---

> ### Author Response · Authors · 2025-11-20
> **Response to reviewer xLwT**
>
> Thank you for your thoughtful and constructive feedback. Please see our reponses below.
>
> > [W1] Autoregressive models can sometimes face the challenge of error accumulation, where an initial mistake in an early token or pixel patch can amplify throughout the rest of the generation.
>
> Thanks for pointing out the issue of error accumulation in autoregressive models. To address this, we employ a progressive training scheme, which is a form of curriculum learning. The training process is divided into stages, controlled by a "progressive stage index". In the initial stages, the model is trained on only the first few patches of the sequence. As training progresses, we gradually increase the sequence length until the model is trained on the full sequence. This incremental approach stabilizes training by allowing the model to first learn to generate short, coherent sequences before tackling the more difficult task of generating long ones. We combine this progressive curriculum with teacher forcing to mitigate error accumulation more effectively.
>
> > [W2] While this approach offers great efficiency and performs well on standard metrics, we may notice that the qualitative fidelity of its reconstructions doesn’t always match that of larger, slower models, such as diffusion, in more complex scenes.
>
> We acknowledge the importance of finding a balance between efficiency and reconstruction quality. A key advantage of VAR lies in its strong performance despite its small size, making it a particularly suitable choice for EEG decoding tasks where both efficiency and reconstruction quality are crucial.
>
> In terms of qualitative fidelity, we observed that while AVDE tends to produce outputs that are less realistic or recognizable when the neural signals are weak, these outputs actually more closely resemble the true stimuli compared to those generated by diffusion-based counterparts. This behavior is advantageous in practice because it implicitly conveys uncertainty to the user, reducing the risk of misplaced trust in inaccurate reconstructions. In contrast, because the diffusion models are typically kept frozen in previous methods, they rely heavily on learned image priors rather than faithfully visualizing features present in the EEG signals. As a result, the models tend to fill in missing details to produce visually plausible outputs that may not accurately reflect the true stimulus, which can be problematic when the generated images mislead users by appearing convincing but incorrect.
>
> It is worth noting that this advantage of AVDE does not necessarily stem from the VAR architecture itself but is more likely attributable to the training pipeline. As described in the manuscript, we fine-tune the image generator directly on EEG–image pairs instead of keeping it fixed. This training strategy preserves a direct mapping between EEG signals and visual outputs, reducing the tendency of the model to “hallucinate” non-existent content.
>
> > [W3] Moreover, the effectiveness of the reconstruction greatly hinges on the pre-trained image tokenizer, which translates images into tokens. If the tokenizer’s performance is lacking, it can limit the model's potential.
>
> We agree that the reconstruction quality indeed depends on the performance of the pre-trained image tokenizer, as it defines the visual vocabulary through which EEG information is expressed. In our case, we employed a tokenizer trained on high-quality, large-scale image datasets, which provides a sufficiently expressive latent space for general visual reconstruction tasks. Nonetheless, we acknowledge that improvements in tokenizer design, such as domain-specific fine-tuning or tokenizers trained jointly with neural decoding objectives, could further enhance reconstruction fidelity. We plan to explore such directions in future work to better align the tokenizer’s representation space with neural signals.

---

> > ### Comment · Reviewer_xLwT · 2025-11-23
> >
> > Thanks for your responses. Kindly list the changes you would be making in the final version based on all the reviews if accepted.

---

> > > ### Author Response · Authors · 2025-11-24
> > >
> > > Thanks for the suggestion. We will post a general response that summarizes the changes.

---

### Author Response · Authors · 2025-11-24

Dear Reviewers,

We sincerely thank you for your thoughtful and constructive feedback. In response, we have conducted the following additional experiments and analyses, and revised the manuscript accordingly:
1. **Quantitative Neuroscientific Analysis (Figure 5)**: We performed a region-to-scale correlation analysis. Results quantitatively confirm that AVDE's intermediate scales align with the biological hierarchy (e.g., early scales correlate with Occipital, mid scales with Temporal/Parietal, late scales with Frontal/Central regions).
2. **New Ablation Studies (Table 4)**: We implemented LaBraM + Diffusion (LDM-4 & DiT-XL/2) baselines trained under the same training settings as LaBraM + VAR. Results show that AVDE outperforms these diffusion baselines in reconstruction fidelity while being more efficient.
3. **Generalization to MEG (Appendix D)**: We extended our evaluation to the THINGS-MEG dataset, demonstrating AVDE's robustness across neural modalities.

We greatly appreciate your time and effort in reviewing our work and look forward to your feedback on the revised manuscript.

---

> ### Comment · Reviewer_xLwT · 2025-11-27
>
> Thanks, increased my rating.

---

> > ### Author Response · Authors · 2025-11-28
> >
> > Thanks again for your engagement and constructive feedback, which helped us strengthen our work.

---

### Meta-Review · Area_Chair_cZkf · 2026-01-07

**Summary:**

1. The paper presents AVDE, which applies autoregressive next-scale prediction (VAR) as an alternative to the conventional multi-stage diffusion process in EEG, departs from existing paradigms in EEG-to-image modeling. Although few new techniques are proposed, this work demonstrates strong technical soundness in the integration of existing components (LaBraM, VAR, VQ-VAE, and CLIP alignment).

2. The proposed method demonstrates remarkable computational advantages, achieving approximately a 90% reduction in parameters while offering significantly faster inference compared to diffusion-based baselines.

3. The model delivers SOTA results in both retrieval and reconstruction tasks. The improvements are consistent across benchmarks, clearly surpassing prior methods and confirming the model’s technical strength.

4. Lightweight and efficient model design positions AVDE as a promising candidate for real-world brain–computer interface (BCI) applications.

5. Reviewers concerned about the absence of certain ablation baselines and the inconsistency in diffusion sampling steps. In response, the authors supplemented the missing experiments, clarified the computational bottlenecks, and provided additional experimental results.

**Reviewer Concerns:**

### Resolved
The authors have successfully addressed the majority of the reviewers’ technical and methodological concerns:
- They clarified the fairness of the efficiency comparison by explaining the computational bottlenecks in diffusion sampling, and they substantially expanded the ablation studies to isolate the contributions of the encoder, generator, and model structure.

- The novelty issue was also clarified — the innovation lies not in introducing new modules but in the integrated training and direct EEG–image alignment strategy, which distinguishes AVDE from traditional multi-stage diffusion frameworks.

- The authors provided quantitative evidence for the cortical–scale correspondence claim through new EEG–image similarity analyses, and they added detailed computational profiling, including FLOPs, inference time, and memory usage.

- They strengthened the empirical robustness of the work by including results for all subjects and explaining the expected mismatch between validation loss and visual reconstruction quality, which is common in generative models.

### Unresolved Concerns

- Theoretical Depth: The authors provided an analysis of the relationship between different scales in AVDE and the corresponding biological hierarchical levels of visual processing. While technically sound, the framework builds on existing architectures and lacks a deeper theoretical innovation.

- Cross-Subject Generalization: Although pre-training improved performance by over 20%, cross-subject accuracy remains limited, suggesting the need for larger and more diverse data.

- Reproducibility: Full external replication has not yet been confirmed.

**Reviewer Scores:**

Reviewer xLwT explicitly indicated that he would raise his score. The other reviewers did not provide explicit follow-ups, but given that most of their concerns have been resolved, it is reasonable to assume they may also increase their scores.

---

### Decision · Program_Chairs · 2026-01-26

Accept (Poster)